# Learning Molecular Chirality via Chiral Determinant Kernels

**Runhan Shi**[1], **Zhicheng Zhang**[1], **Letian Chen**[1,2], **Gufeng Yu**[1], **Yang Yang**[1*]

[1]AGI Institute, School of Computer Science, Shanghai Jiao Tong University
[2]Shanghai Innovation Institute
`{han.run.jiangming, zhicheng.zhang, clt2001, jm5820zz}@sjtu.edu.cn`
`yangyang@cs.sjtu.edu.cn`

## Abstract

Chirality is a fundamental molecular property that governs stereospecific behavior in chemistry and biology. Capturing chirality in machine learning models remains challenging due to the geometric complexity of stereochemical relationships and the limitations of traditional molecular representations that often lack explicit stereochemical encoding. Existing approaches to chiral molecular representation primarily focus on central chirality, relying on handcrafted stereochemical tags or limited 3D encodings, and thus fail to generalize to more complex forms such as axial chirality. In this work, we introduce **ChiDeK** (**Chi**ral **De**terminant **K**ernels), a framework that systematically integrates stereogenic information into molecular representation learning. We propose the chiral determinant kernel to encode the SE(3)-invariant chirality matrix and employ cross-attention to integrate stereochemical information from local chiral centers into the global molecular representation. This design enables explicit modeling of chiral-related features within a unified architecture, capable of jointly encoding central and axial chirality. To support the evaluation of axial chirality, we construct a new benchmark for electronic circular dichroism (ECD) and optical rotation (OR) prediction. Across four tasks, including R/S configuration classification, enantiomer ranking, ECD spectrum prediction, and OR prediction, ChiDeK achieves substantial improvements over state-of-the-art baselines, most notably yielding over 7% higher accuracy on axially chiral tasks on average.

## 1 Introduction

Chirality, the property of molecules that are non-superimposable mirror images, is a fundamental concept in chemistry, biology, and materials science (Moss, 1996). It plays a central role in shaping molecular behavior across diverse applications, including biomolecular recognition (Ceramella et al., 2022), stereoselective synthesis (Gaucherand et al., 2024), and enantioselective catalysis (He et al., 2023). Chiral molecules, often existing as pairs of enantiomers, can exhibit drastically different chemical and biological properties despite sharing identical compositions (Peluso & Chankvetadze, 2022). Such differences are especially critical in drug efficacy (Liu et al., 2023b), toxicity (McVicker & O'Boyle, 2024), and protein-binding affinity (Schneider et al., 2018; Shen et al., 2013), underscoring the paramount importance of accurately modeling stereochemical information in molecular representation learning.

Despite this significance, capturing chirality in machine learning models remains challenging. Existing molecular representation learning (MRL) approaches predominantly focus on central chirality and often rely on implicit encoding strategies that fail to extract explicit stereochemical features. While recent advances in SE(3)-invariant architectures have shown promise through the incorporation of torsion angles (Klicpera et al., 2021; Liu et al., 2022) and explicitly chirality-aware designs (Adams et al., 2022; Yan et al., 2025), research on comprehensive chirality modeling within MRL remains limited. More complex stereogenic forms, such as axial, planar, and helical chirality, are largely unaddressed, representing a significant gap in current methodologies.

---

*Corresponding author.

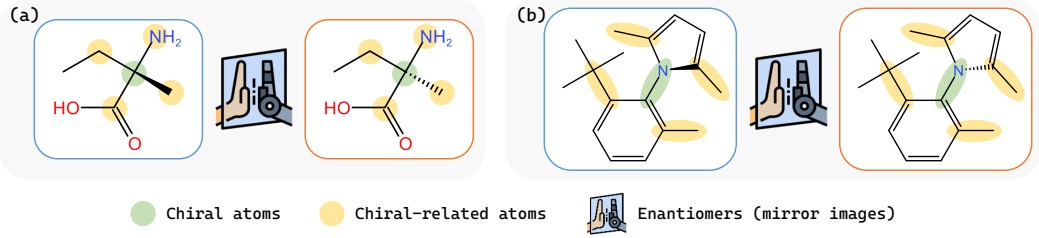

Figure 1: Examples of central chirality (a) and axial chirality (b). The configuration of the chiral atoms (in green) is determined by the spatial arrangement of chiral-related atoms (in orange).

Among the various forms of molecular chirality, central chirality and axial chirality represent the two most prevalent categories. **Central chirality** arises when a tetrahedral atom (typically carbon) is bonded to four distinct substituents. Such an atom constitutes a *chiral center*, and its stereochemical configuration (R or S) (Cahn et al., 1966; Favre & Powell, 2013) is determined by the spatial arrangement of its substituents, as illustrated in Figure 1(a). **Axial chirality** typically results from restricted rotation around a chemical axis, often a bond connecting two aromatic or aliphatic groups. Here, chirality (Ra/M or Sa/P) is determined by the relative arrangement of substituent groups around the stereogenic axis, as shown in Figure 1(b).

Existing computational approaches encounter fundamental limitations in capturing chirality. E(3)-invariant architectures that rely exclusively on pairwise distances or bond angles, such as SchNet (Schütt et al., 2017) and DimeNet (Gasteiger et al., 2020b;a), are inherently incapable of distinguishing enantiomers since these geometric quantities remain unchanged under reflection. While SE(3)-invariant models, including SphereNet (Liu et al., 2022), ChIRo (Adams et al., 2022), and ChiGNN (Yan et al., 2025), address this limitation through directional information and equivariant features, they face critical challenges. Most notably, molecular graphs are typically treated as homogeneous structures where chiral atoms are not explicitly differentiated, causing stereochemical signals to be diluted during global aggregation. Recent approaches like ChiGNN attempt to incorporate chirality through permutation-resolution strategies, but they still fail to extract expressive, localized descriptors that faithfully characterize chiral centers or stereogenic axes. These constraints highlight the urgent need for a unified framework capable of systematically modeling diverse chirality types while explicitly encoding stereochemical features that govern molecular behavior.

To address the above limitations, we propose the **ChiDeK** model (**Chi**ral **De**terminant **K**ernels), which learns molecular chirality based on the determinant of the chirality matrix and applies cross-attention between chiral and non-chiral atoms to explicitly capture stereogenic information. The introduced chiral determinant kernels embed the SE(3)-invariant chirality matrix (Shi et al., 2026) for each chiral atom, capturing stereogenic information for both central and axial configurations. A cross-attention mechanism enhanced by pair representations then propagates the influence of chiral atoms (queries) with chiral-related and non-chiral atoms (keys and values) into the global molecular representation, enhancing stereochemical sensitivity. Our main contributions are:

- We introduce ChiDeK, a unified architecture that systematically encodes both central and axial chirality, and evaluate it comprehensively on multiple chirality-aware prediction tasks.

- We design the chiral determinant kernel for chiral atoms that embeds the SE(3)-invariant chirality matrix, effectively capturing stereogenic features for chirality.

- We construct and release a dataset of axially chiral molecules for the prediction of electronic circular dichroism (ECD) and optical rotation (OR), providing a benchmark for this underexplored stereogenic type.

- Across multiple tasks, including R/S classification, enantiomer ranking, ECD prediction, and OR prediction, ChiDeK demonstrates substantial improvements over state-of-the-art baselines, particularly for the axial chirality task.

## 2 RELATED WORK

**Learning 3D Representations of Molecules.** Recent advances in molecular machine learning have heavily leveraged 3D geometric message passing and attention architectures, where interatomic interactions are modeled via internal coordinates such as distances, angles, and torsions. SchNet (Schütt et al., 2017), a pioneering architecture for predicting quantum mechanical properties, parameterizes messages using radial basis expansions of pairwise distances. DimeNet (Gasteiger et al., 2020b) and DimeNet++ (Gasteiger et al., 2020a) extend this framework by incorporating angular features through spherical Bessel functions, while DeepH-E3 (Gong et al., 2023) learns Hamiltonians from density functional theory (DFT) calculations. BOTNet (Batatia et al., 2025) further simplifies neural equivariant interatomic potentials (NequIP) (Batzner et al., 2022), a high-performing equivariant message passing framework. These models are designed to be E(3)-invariant, which are robust to translations, rotations, and reflections. While this invariance is desirable for many property prediction tasks, it enforces reflection-symmetry, rendering such models unable to distinguish enantiomers that differ only by mirror inversion (Satorras et al., 2021; Dumitrescu et al., 2025).

To remedy this shortcoming, SE(3)-invariant architectures have been developed, which maintain sensitivity to reflections. Tensor Field Networks (Thomas et al., 2018) and SE(3)-Transformers (Fuchs et al., 2020) achieve equivariance through the use of spherical harmonics and irreducible representations. Path-MPNN (Flam-Shepherd et al., 2021), GemNet (Klicpera et al., 2021), and SphereNet (Liu et al., 2022) further enhance 3D GNNs by embedding angular and torsional information. More recently, MoleculeSED (Liu et al., 2023a) introduces reflection-sensitive stochastic differential equations for molecular modeling.

**Explicit Representation Learning of Molecular Chirality.** A number of studies explicitly incorporate stereochemical information into molecular representations. Early efforts introduce handcrafted descriptors such as the chirality code (Aires-de Sousa et al., 2004), physicochemical atomic stereo (PAS) descriptors (Zhang & Aires-de Sousa, 2006), relative chirality indices (RCI) (Natarajan et al., 2007), and chiral cliffs (Schneider et al., 2018). More recently, MAP4C (Orsi & Reymond, 2024) extends molecular fingerprints to capture stereochemical tags directly. Linear notations such as SMILES (Weininger, 1988), SELFIES (Krenn et al., 2022), and fragSMILES (Mastrolorito et al., 2025) encode molecules as sequences that can include chirality tags, enabling models like Transformers (Vaswani et al., 2017) to process stereochemical information (Yoshikai et al., 2024; Yang et al., 2025). However, these tags serve only as symbolic indicators without explicit 3D structural context, making the resulting representations less informative for capturing stereogenic geometry. Graph neural networks have also been adapted to represent chirality more explicitly. Tetra-DMPNN (Pattanaik et al., 2020) modifies message passing in 2D GNNs with asymmetric updates to encode tetrahedral chirality. SPMS (Xu et al., 2021) introduces spherical projection descriptors with convolutional architectures for stereostructure modeling. ChIRo (Adams et al., 2022) learns 3D representations invariant to bond rotations yet sensitive to stereoisomers. MolKGNN (Liu et al., 2023c) leverages the tetrahedron volume calculation to capture the neighbor ordering for chirality. CFFN (Du et al., 2023) combines 2D graph topology and 3D geometry to build stereochemistry-aware molecular representations. GCPNet (Morehead & Cheng, 2024) develops geometric representations tailored for biomolecular graphs. ChiENN (Gaiński et al., 2023) and ChiGNN (Yan et al., 2025) explicitly consider atom permutations related to chirality to improve stereochemical discrimination. ChiralCat (Peng et al., 2025) integrates 3D molecular structures with LLM-generated textual descriptions to perform chiral type classification.

Despite these advances, the vast majority of prior work focuses primarily on central chirality. Other stereogenic forms, such as axial chirality, remain largely underexplored, limiting current models' ability to generalize across the full spectrum of stereochemistry. Although methods such as SPMS are theoretically capable of representing additional forms of chirality, they have not been systematically evaluated in these settings, leaving their practical effectiveness uncertain.

## 3 PRELIMINARIES

### 3.1 CHIRAL AND CHIRAL-RELATED ATOMS

To construct a meaningful molecular representation that explicitly encodes chirality, we distinguish between *chiral atoms* and their corresponding *chiral-related atoms* (Figure 1). Given a molecule $z$

with $M$ atoms, where the index set of atoms is $\mathcal{I} = \{1, \ldots, M\}$, we define three disjoint subsets :

$$\mathcal{I}_c, \quad \mathcal{I}_r = \bigcup_{i \in \mathcal{I}_c} \{j : j \in \mathcal{N}(i)\} \setminus \mathcal{I}_c, \quad \mathcal{I}_n = \mathcal{I} \setminus (\mathcal{I}_c \cup \mathcal{I}_r), \tag{1}$$

where $\mathcal{I}_c$, $\mathcal{I}_r$, and $\mathcal{I}_n \subset \mathcal{I}$ are the index sets of chiral, chiral-related, and non-chiral atoms, respectively, and $\mathcal{N}(i)$ denotes the index set of substituents around a chiral atom $i$.

For central chirality, a chiral atom $i \in \mathcal{I}_c$ typically has four different substituent groups. The corresponding four neighboring atoms define its chiral-related set $\mathcal{N}(i)$. For axial chirality, chirality arises along a stereogenic axis (e.g., in biaryls), where steric hindrance prevents free rotation. In this case, the substituent groups are partitioned into four distinct sets around the axis, and we select the nearest atom from each group as the chiral-related atoms.

## 3.2 CHIRALITY MATRIX

To explicitly encode stereogenic information, we adopt the chirality matrix introduced in ChiralFinder (Shi et al., 2026). Formally, for a chiral atom $i \in \mathcal{I}_c$ with ordered substituents $\mathcal{N}(i) = (r_{i,1}, r_{i,2}, r_{i,3}, r_{i,4})$, the *chirality matrix* is defined as

$$\boldsymbol{M}_{\mathrm{C}}(i) := \left( (\boldsymbol{x}_{r_{i,1}} - \boldsymbol{x}_i) \quad (\boldsymbol{x}_{r_{i,2}} - \boldsymbol{x}_i) \quad (\boldsymbol{x}_{r_{i,4}} - \boldsymbol{x}_{r_{i,3}}) \right)^{\top}, \tag{2}$$

where $\boldsymbol{x}_i \in \mathbb{R}^3$ denotes the 3D position of the chiral atom $i$, and $\boldsymbol{x}_{r_{i,j}}$ denotes the position of its $j$-th substituent. The corresponding *chirality product* is then defined as

$$P_{\mathrm{C}}(i) := \left( (\boldsymbol{x}_{r_{i,1}} - \boldsymbol{x}_i) \times (\boldsymbol{x}_{r_{i,2}} - \boldsymbol{x}_i) \right) \cdot (\boldsymbol{x}_{r_{i,4}} - \boldsymbol{x}_{r_{i,3}}) = \det \left( \boldsymbol{M}_{\mathrm{C}}(i) \right), \tag{3}$$

where $\times$ denotes the vector cross product, $\cdot$ denotes the dot product, and $\det(\cdot)$ is the determinant of a given matrix. Appendix D.1 provides details on how $\mathcal{N}(i)$ is determined.

**Proposition 3.1.** *Given a chiral atom $i$, the chirality product $P_{\mathrm{C}}(i)$ is invariant under rigid-body translation and rotation, and changes sign under reflection:*

$$\det \left( \boldsymbol{M}_{\mathrm{C}}(i) \right) = \det \left( R_1 \boldsymbol{M}_{\mathrm{C}}(i) \right), \quad \det \left( R_2 \boldsymbol{M}_{\mathrm{C}}(i) \right) = -\det \left( \boldsymbol{M}_{\mathrm{C}}(i) \right), \tag{4}$$

*where $R_1 \in \mathrm{SE}(3)$ denotes any rigid-body motion with rotation $r \in \mathrm{SO}(3)$ and translation $t \in \mathbb{R}^3$, and $R_2 \in O^-(3)$ denotes any reflection (orthogonal transformation with determinant $-1$).*

**Lemma 3.1.** *Let $(\boldsymbol{z}_1, \boldsymbol{z}_2)$ be a pair of enantiomers differing only in the stereochemical configuration of atom $i$. Then*

$$configuration(i) = \begin{cases} R, & \det(\boldsymbol{M}_{\mathrm{C}}(i)) > 0, \\ S, & \det(\boldsymbol{M}_{\mathrm{C}}(i)) < 0. \end{cases} \tag{5}$$

Proof of Lemma 3.1 is provided in Appendix F.1.

## 4 METHOD

### 4.1 CHIDEK FRAMEWORK

The proposed ChiDeK framework, illustrated in Figure 2(a), is designed to capture stereochemical information by explicitly modeling chiral atoms through chiral determinant kernels. Given a molecule $\boldsymbol{z} = (\boldsymbol{X}, \boldsymbol{H})$, where $\boldsymbol{X} = (\boldsymbol{x}_1, \ldots, \boldsymbol{x}_M) \in \mathbb{R}^{3M}$ specifies the 3D coordinates of the $M$ atoms, and $\boldsymbol{H} = (\boldsymbol{h}_1, \ldots, \boldsymbol{h}_M) \in \mathbb{R}^{dM}$ encodes atom-level features, we construct a molecular graph $\mathcal{G} = (\mathcal{V}, \mathcal{E})$, where the node set $\mathcal{V}$ is partitioned into chiral atoms, chiral-related atoms, and non-chiral atoms, and the edge set $\mathcal{E}$ incorporates geometric distances between atoms.

The forward pass of ChiDeK proceeds in three stages: (1) a *chiral encoder* computes chiral-sensitive embeddings for chiral atoms via chiral determinant kernels, while three separate multi-layer projectors compute chiral-invariant embeddings for chiral, chiral-related, and non-chiral atoms; (2) the multi-layer *chiral cross-attention* mechanism refines embeddings of chiral atoms (queries) by attending to heterogeneous neighbors (keys/values), guided by distance-aware Gaussian Kernel with Pair Type (GKPT) biases; (3) the resulting chiral-aware representations are aggregated and passed through task-specific heads (e.g., chirality classification, spectrum regression).

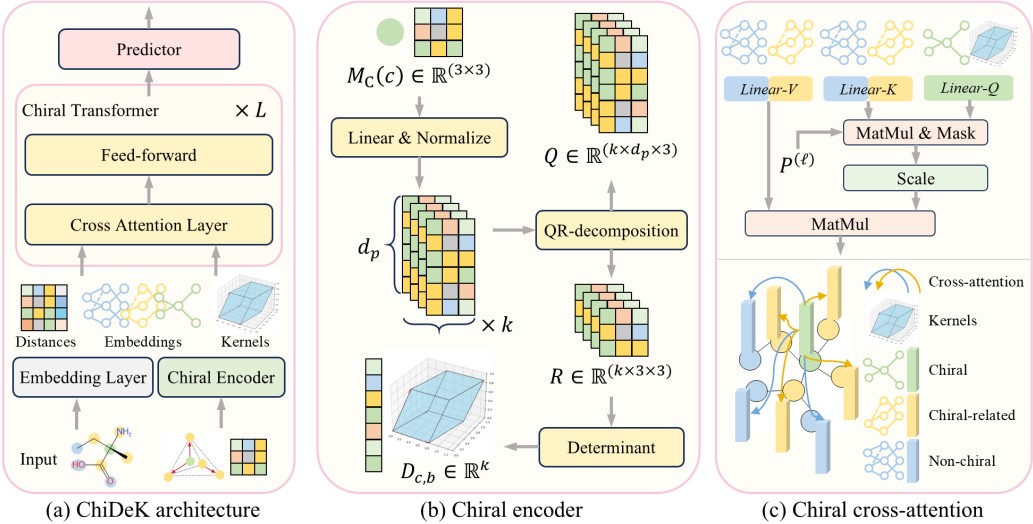

Figure 2: Overview of the ChiDeK architecture (a). It consists of a chiral encoder (b), a chiral transformer incorporating $L$ cross attention layers (c), and a predictor for predicting chiral properties.

## 4.2 CHIRAL ENCODER

A key requirement for learning molecular chirality is to design features that are sensitive to reflections but invariant to global translations and rotations. The classical determinant of a $3 \times 3$ chirality matrix $\boldsymbol{M}_{\mathrm{C}}(i)$ provides exactly such a property, as it encodes the signed volume spanned by three bond vectors around a chiral atom. To generalize this construction into a high-dimensional latent space, we introduce the chiral encoder based on chiral determinant kernels.

**Chiral Determinant Kernel.** To explicitly represent chiral atoms with stereogenic information, we introduce the chiral determinant kernel, a determinant-based kernel that encodes the oriented volume spanned by triplets of projected atomic vectors with QR decomposition.

Let $\boldsymbol{M}_{\mathrm{C}} \in \mathbb{R}^{B \times 3 \times 3}$ denote the batch of chirality matrices and $\boldsymbol{W} \in \mathbb{R}^{k \times d_p \times 3}$ the learnable weights, where $k$ is the number of kernels and $d_p$ is the projection dimension. After broadcasting to $\boldsymbol{M}'_{\mathrm{C}} \in \mathbb{R}^{B \times k \times 3 \times 3}$ and $\boldsymbol{W}' \in \mathbb{R}^{B \times k \times d_p \times 3}$, we compute

$$\boldsymbol{O}_{b,k} = \boldsymbol{W}'_k \, \boldsymbol{M}'_{\mathrm{C},b}, \quad \boldsymbol{O} \in \mathbb{R}^{B \times k \times d_p \times 3}, \tag{6}$$

where $\boldsymbol{M}'_{\mathrm{C},b} \in \mathbb{R}^{3 \times 3}$ is the $b$-th chirality matrix, $\boldsymbol{W}'_k \in \mathbb{R}^{d_p \times 3}$ is the $k$-th kernel, and multiplication is standard matrix multiplication along the last two dimensions. Then, we reshape the output $\boldsymbol{O}$ from $\mathbb{R}^{B \times k \times d_p \times 3}$ to $\boldsymbol{O}' \in \mathbb{R}^{(BK) \times d_p \times 3}$. Applying layer normalization along the $d_p$ dimension yields

$$\tilde{\boldsymbol{O}} = \mathrm{LayerNorm}\left(\boldsymbol{O}'\right) \in \mathbb{R}^{(BK) \times d_p \times 3}. \tag{7}$$

Next, we perform a QR decomposition for each slice:

$$\tilde{\boldsymbol{O}}_{(bk)} = \boldsymbol{Q}_{(bk)} \boldsymbol{R}_{(bk)}, \quad \boldsymbol{Q}_{(bk)}^{\top} \boldsymbol{Q}_{(bk)} = \boldsymbol{I}_3, \quad \boldsymbol{Q}_{(bk)} \in \mathbb{R}^{d_p \times 3}, \; \boldsymbol{R}_{(bk)} \in \mathbb{R}^{3 \times 3}, \tag{8}$$

for all $b \in \{1, \dots, B\}$ and $k \in \{1, \dots, k\}$. We then compute the determinant of each factor:

$$\boldsymbol{D}_{c,b,k} = \det\left(\boldsymbol{R}_{(bk)}\right), \quad \boldsymbol{D}_c \in \mathbb{R}^{B \times k}. \tag{9}$$

Thus, each chiral atom is embedded into a $k$-dimensional representation that generalizes the determinant, provides differentiability (Roberts & Roberts, 2020), and explicitly encodes stereochemistry. Then the initial hidden representations of chiral, chiral-related, and non-chiral atoms are

$$\boldsymbol{H}_c = \boldsymbol{D}_c + f_c\left(\boldsymbol{H}_{\mathcal{I}_c}\right), \quad \boldsymbol{H}_r = f_r\left(\boldsymbol{H}_{\mathcal{I}_r}\right), \quad \boldsymbol{H}_n = f_n\left(\boldsymbol{H}_{\mathcal{I}_n}\right), \tag{10}$$

where $f_c, f_r, f_n$ are embedding layers for atomic features. We finally prepend a learnable global chiral token before chiral atoms to enhance chiral representation.

### 4.3 CHIRAL TRANSFORMER

**Gaussian kernel with pair type (GKPT).** Distances between atoms provide crucial stereochemical cues. To encode them, we adopt GKPT (Shuaibi et al., 2021; Zhou et al., 2023). Formally, for input distance $x$ and pair type $e$, the GKPT is defined as

$$\text{GKPT}\left((x, e), \boldsymbol{\mu}, \boldsymbol{\sigma}\right) = \mathcal{G}\left(\mathbf{E}_1(e) \cdot x + \mathbf{E}_2(e), \boldsymbol{\mu}, \boldsymbol{\sigma}\right), \tag{11}$$

where $\mathcal{G}\left(x', \boldsymbol{\mu}, \boldsymbol{\sigma}\right) = \frac{1}{\sqrt{2\pi}\boldsymbol{\sigma}} \exp\left(-\frac{1}{2}\left(\frac{x'-\boldsymbol{\mu}}{\boldsymbol{\sigma}}\right)^2\right)$, $\mathbf{E}_1, \mathbf{E}_2 \in \mathbb{R}^{N_e \times k}$ are learnable embeddings, $N_e$ is the number of pair types, and $\boldsymbol{\mu}, \boldsymbol{\sigma} \in \mathbb{R}^k$ parameterize the Gaussian kernels. Here, we distinguish between two types of edges: chiral atoms with chiral-related atoms and with non-chiral atoms. The GKPT module is applied to geometric distance $x_{ij}$ with edge type $e_{ij}$, between atom $i \in \mathcal{I}_c$ and $j \in \mathcal{I}_r \bigcup \mathcal{I}_n$, producing the bias term

$$p_{ij}^{(0)} = \text{GKPT}\left((x_{ij}, e_{ij}), \boldsymbol{\mu}, \boldsymbol{\sigma}\right) \boldsymbol{w}_p, \tag{12}$$

with learnable projection $\boldsymbol{w}_p$. The resulting pairwise bias matrix is

$$\boldsymbol{P}^{(0)} = \left[p_{ij}^{(0)}\right]_{1 \leq i \leq |\mathcal{I}_c|,\ 1 \leq j \leq |\mathcal{I}_r| + |\mathcal{I}_n|} \in \mathbb{R}^{|\mathcal{I}_c| \times (|\mathcal{I}_r| + |\mathcal{I}_n|)}. \tag{13}$$

This pairwise representation encodes chemically relevant interactions, informing the subsequent attention computation and enabling the model to learn complex spatial dependencies.

**Chiral Cross-attention.** We introduce a cross-attention mechanism enhanced by pairwise representations, where chiral atoms serve as queries, while chiral-related and non-chiral atoms provide keys and values. Let $\boldsymbol{H}_c$, $\boldsymbol{H}_r$, and $\boldsymbol{H}_n$ denote the hidden representations of chiral atoms $\mathcal{I}_c$, chiral-related atoms $\mathcal{I}_r$, and non-chiral atoms $\mathcal{I}_n$, respectively. At each layer $\ell$ (totally $L$ layers), the linear projections for keys and values are separated according to atom types and defined as

$$\boldsymbol{K}_r^{(\ell)} = \boldsymbol{H}_r W_{K_r}^{(\ell)\top}, \quad \boldsymbol{V}_r^{(\ell)} = \boldsymbol{H}_r W_{V_r}^{(\ell)\top}, \quad \boldsymbol{K}_n^{(\ell)} = \boldsymbol{H}_n W_{K_n}^{(\ell)\top}, \quad \boldsymbol{V}_n^{(\ell)} = \boldsymbol{H}_n W_{V_n}^{(\ell)\top}, \tag{14}$$

where $W_{K_r}^{(\ell)}, W_{V_r}^{(\ell)}, W_{K_n}^{(\ell)}$, and $W_{V_n}^{(\ell)} \in \mathbb{R}^{k \times k}$ are learnable projection matrices. Let $\boldsymbol{H}_c^{(\ell)}$ denotes the output at layer $\ell$, the resulting queries, keys, and values are

$$\boldsymbol{Q}^{(\ell)} = \boldsymbol{H}_c^{(\ell-1)} W_Q^{(\ell)\top}, \quad \boldsymbol{K}^{(\ell)} = \left[\boldsymbol{K}_r^{(\ell)},\ \boldsymbol{K}_n^{(\ell)}\right], \quad \boldsymbol{V}^{(\ell)} = \left[\boldsymbol{V}_r^{(\ell)},\ \boldsymbol{V}_n^{(\ell)}\right], \tag{15}$$

where $W_Q^{(\ell)} \in \mathbb{R}^{k \times k}$ is the learnable query projection, $\boldsymbol{H}_c^{(0)} = \boldsymbol{H}_c$, and $[\cdot, \cdot]$ denotes concatenation. At each layer $\ell$, we maintain a learnable pairwise bias $p_{ij}^{(\ell)}$ between atom $i$ (chiral) and atom $j$ (chiral-related or non-chiral), updated via the query-key interaction. See Appendix C for details.

**Predictor.** We utilize a lightweight predictor to predict chirality properties. It consists of two linear layers with GELU activation Hendrycks & Gimpel (2016). This simple yet effective architecture maps the final global representation to the target prediction space.

### 4.4 THEORETICAL ANALYSIS FOR CHIRAL ENCODER

The chirality product obtained from the chirality matrix (Shi et al., 2026) exhibits inherent sensitivity to stereochemical variation. We demonstrate that a chiral determinant kernel maintains this desirable characteristic.

**Lemma 4.1.** *Let $\boldsymbol{M}_{\mathrm{C}}(i) \in \mathbb{R}^{3 \times 3}$ be the chirality matrix of atom $i$, and let $\boldsymbol{W} \in \mathbb{R}^{d_p \times 3}$ be a full column-rank projection. Define QR decomposition of $\boldsymbol{O} = \boldsymbol{W} \boldsymbol{M}_{\mathrm{C}}(i) \in \mathbb{R}^{d_p \times 3}$ as*

$$\boldsymbol{O} = \boldsymbol{Q}\boldsymbol{R}, \quad \boldsymbol{Q}^\top \boldsymbol{Q} = \boldsymbol{I}_3, \quad \boldsymbol{Q} \in \mathbb{R}^{d_p \times 3}, \quad \boldsymbol{R} \in \mathbb{R}^{3 \times 3}. \tag{16}$$

*Then the generalized chirality product $P_{\boldsymbol{W}}(i) = \det(\boldsymbol{R})$ satisfies*

$$P_{\boldsymbol{W}}(i) = \alpha(\boldsymbol{W}) \cdot P_{\mathrm{C}}(i), \tag{17}$$

*where $P_{\mathrm{C}}(i)$ is the chirality product, and $\alpha(\boldsymbol{W}) > 0$ is a scaling factor depending only on $\boldsymbol{W}$ but not on the atom coordinates. $P_{\boldsymbol{W}}(i)$ satisfies Proposition 3.1 and Lemma 3.1.*

The proof of Lemma 4.1 is provided in Appendix F.2.

**Discussion on rank deficiency.** If $W$ is not full column-rank, then $O$ becomes rank-deficient and the determinant $\det(R)$ degenerates to $0$, resulting in the loss of chirality information. While the magnitude of $\det(R)$ depends on $\alpha(W)$, the sign remains a reliable indicator of chiral configurations as long as $W$ has full column rank. Hence, ensuring rank sufficiency of $W$ is critical for stability and discriminative power. To guarantee full column rank of $W$, we adopt two strategies in practice:

**Weight regularization.** We introduce a penalty term on the weight as part of the loss function:

$$\mathcal{L}_{\text{reg}} = ||W^\top W - I_3||^2, \tag{18}$$

encouraging $W$ to preserve rank during training.

**Auxiliary QR on weights.** Alternatively, we apply QR decomposition directly on $W$ before projection, ensuring that projected neighbor vectors remain linearly independent.

## 5 EXPERIMENTS

### 5.1 MAIN EXPERIMENTS

**Datasets.** We evaluate our method on both centrally and axially chiral molecules using multiple benchmarks, as shown in Appendix Table 4. R/S classification and enantiomer ranking datasets for central chirality are from ChIRo (Adams et al., 2022). ECD prediction for central chirality is from CMCDS (Li et al., 2025). ECD and OR prediction for axial chirality are constructed by ourselves, named ACMP (axial chiral molecular properties). See Appendix A for details about dataset processing, construction, and splits.

**Baselines.** We benchmark ChiDeK against both an E(3)-invariant baseline, DimeNet++ (Gasteiger et al., 2020a), and several SE(3)-invariant baselines, including SphereNet (Liu et al., 2022), ChIRo (Adams et al., 2022), ECDFormer (Li et al., 2025), and ChiGNN (Yan et al., 2025). We also consider the 2D chiral GNN Tetra-DMPNN with permutation (p) and concatenation (c) variants (Pattanaik et al., 2020), and the stereostructure descriptor-based SPMS Xu et al. (2021).

**Implementation Details.** All experiments are implemented in PyTorch (Paszke et al., 2019) and conducted on NVIDIA RTX 3090 GPUs. See Appendix D for more training details.

#### 5.1.1 EXPERIMENTAL RESULTS FOR CENTRAL CHIRALITY

Table 1: R/S classification, enantiomer ranking, and ECD prediction results for central chirality. **Best** and second-best are marked.

| Method | R/S (%) | Ranking (%) | Position | Number | Symbol (%) |
|---|---|---|---|---|---|
| | **Acc ↑** | **Acc ↑** | **RMSE ↓** | **RMSE ↓** | **Acc ↑** |
| DimeNet++ | $65.7 \pm 2.9$ | $58.4 \pm 0.2$ | $2.12 \pm 0.19$ | $1.04 \pm 0.15$ | $50.8 \pm 0.1$ |
| Tetra-DMPNN (c) | $99.7 \pm 0.1$ | $70.1 \pm 0.5$ | $2.38 \pm 0.12$ | $1.25 \pm 0.09$ | $50.0 \pm 0.1$ |
| Tetra-DMPNN (p) | $\underline{99.7 \pm 0.1}$ | $67.6 \pm 0.6$ | $2.36 \pm 0.16$ | $1.28 \pm 0.08$ | $50.0 \pm 0.1$ |
| SphereNet | $98.2 \pm 0.2$ | $68.6 \pm 0.3$ | $2.36 \pm 0.15$ | $\underline{1.02 \pm 0.11}$ | $\underline{51.9 \pm 0.3}$ |
| ChIRo | $98.5 \pm 0.2$ | $\underline{72.0 \pm 0.5}$ | $2.67 \pm 0.13$ | $1.22 \pm 0.13$ | $51.0 \pm 0.4$ |
| ECDFormer | $92.3 \pm 1.2$ | $58.6 \pm 0.3$ | $\mathbf{2.02 \pm 0.10}$ | $\mathbf{1.01 \pm 0.09}$ | $50.3 \pm 0.3$ |
| ChiGNN | $84.5 \pm 0.9$ | $59.6 \pm 0.4$ | $2.39 \pm 0.18$ | $1.24 \pm 0.10$ | $49.9 \pm 0.5$ |
| SPMS | $81.4 \pm 0.7$ | $60.4 \pm 0.4$ | $2.58 \pm 0.17$ | $1.22 \pm 0.12$ | $50.9 \pm 0.3$ |
| ChiDeK (Ours) | $\mathbf{99.8 \pm 0.1}$ | $\mathbf{72.8 \pm 0.2}$ | $2.20 \pm 0.14$ | $1.18 \pm 0.09$ | $\mathbf{53.3 \pm 0.6}$ |

**R/S Classification.** Table 1 reports the results. ChiDeK achieves the best performance close to 100%. This clearly demonstrates the effectiveness of our chirality-aware design. In contrast, the E(3)-invariant DimeNet++ completely fails on this task, which is as expected. We present a visualization of cross attention weights in Appendix G. Note that, as emphasized in prior work (Adams

et al., 2022), this R/S classification task is necessary but not sufficient for meaningful chiral learning. Nonetheless, strong performance on this benchmark is an essential prerequisite for chirality-sensitive applications.

**Enantiomer Ranking.** Table 1 demonstrates that ChiDeK achieves optimal performance, surpassing the second-best method ChIRo by 0.8%. The modest performance gap between ChiDeK and ChIRo can be attributed to the limited structural diversity of the dataset, as all molecules contain only a single chiral center.

**ECD Spectrum Prediction.** Since the number and positions of peaks remain invariant across enantiomers while the symbols of peak intensities are inverted, accurately predicting peak heights represents the most critical aspect of ECD spectrum modeling. A model that successfully recovers peak counts and positions but fails to distinguish peak height symbols is ineffective and impractical. Consequently, we require a model that jointly predicts all three components, providing comprehensive ECD spectrum prediction. As shown in Table 1, ChiDeK achieves optimal performance in predicting peak height symbols while maintaining comparable accuracy for peak positions and counts.

However, the sign accuracy remains only slightly above 50% across all models. We hypothesize that this is largely due to *dataset inconsistencies*: the original CMCDS dataset provides RDKit-generated (Landrum, 2025) conformers, whereas the ECD labels are computed from optimized geometries. This mismatch causes poor performance. This underscores the inherent difficulty of reliable ECD spectrum modeling and suggests that consistent datasets are essential for more robust evaluation.

### 5.1.2 EXPERIMENTAL RESULTS FOR AXIAL CHIRALITY

Table 2: OR and ECD prediction results for axial chirality. **Best** and second-best are marked.

| Method | Rotation (%) | Position | Number | Symbol (%) |
|---|---|---|---|---|
| | Acc ↑ | RMSE ↓ | RMSE ↓ | Acc ↑ |
| DimeNet++ | $50.0 \pm 0.0$ | $3.62 \pm 0.17$ | $1.13 \pm 0.08$ | $50.0 \pm 0.1$ |
| Tetra-DMPNN (c) | $50.0 \pm 0.1$ | $3.15 \pm 0.12$ | $1.20 \pm 0.15$ | $52.8 \pm 0.2$ |
| Tetra-DMPNN (p) | $50.0 \pm 0.1$ | $3.16 \pm 0.10$ | **$1.05 \pm 0.12$** | $52.8 \pm 0.2$ |
| SphereNet | $52.5 \pm 0.2$ | $3.36 \pm 0.22$ | $1.08 \pm 0.09$ | $52.4 \pm 0.4$ |
| ChIRo | $50.0 \pm 0.1$ | $3.78 \pm 0.18$ | $1.11 \pm 0.11$ | $51.1 \pm 0.3$ |
| ECDFormer | $53.5 \pm 0.3$ | $3.89 \pm 0.25$ | $1.16 \pm 0.14$ | $51.5 \pm 0.5$ |
| ChiGNN | $50.2 \pm 0.1$ | **$2.89 \pm 0.12$** | $1.06 \pm 0.13$ | $50.8 \pm 0.8$ |
| SPMS | $65.0 \pm 0.6$ | $3.69 \pm 0.21$ | $1.16 \pm 0.15$ | $60.4 \pm 0.3$ |
| ChiDeK (Ours) | **$69.2 \pm 0.5$** | $3.24 \pm 0.14$ | **$1.05 \pm 0.12$** | **$71.2 \pm 0.6$** |

**OR Prediction.** Table 2 shows that ChiDeK delivers the strongest performance, exceeding the second-best method by a noticeable 4.2% margin. Notably, all baseline models, except SPMS, achieve accuracy below 55%, indicating their inability to capture axial chirality due to their neglect of stereogenic axes. SPMS, which incorporates spherical stereostructural projections, can partially encode such information. These findings underscore the inherent complexity of the task and highlight the limitations of previous models in accurately representing complex chirality.

**ECD Spectrum Prediction.** Table 2 reports the results on the axial chirality ECD prediction task. ChiDeK achieves the highest accuracy in predicting the symbols of peak heights, exceeding the second-best model by a substantial margin of 10.8%. It also delivers competitive performance in predicting peak positions and numbers. Similar to OR prediction, SPMS exhibits limited performance, while all other baseline models achieve less than 55% accuracy in symbol prediction. For fine-grained analysis, we present performance breakdown across seven axial-chirality subtypes in Appendix B.1. Additionally, we evaluate general molecular property prediction tasks in Appendix B.2 to confirm that our model maintains full expressive power, where chirality is less critical.

Figure 3 presents a representative prediction comparison. ChiDeK successfully assigns opposite peak-height symbols to opposite stereochemical configurations, thereby correctly distinguishing axial enantiomers. In contrast, ChiGNN and Tetra-DMPNN (permute) produce identical peak-height

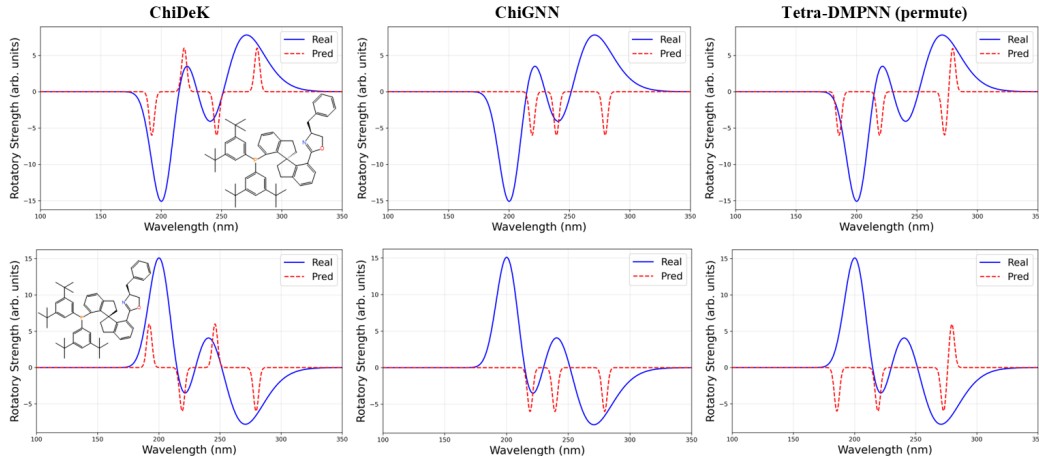

Figure 3: An axial ECD prediction example of a pair of enantiomers. Each row shows the predictions for one configuration across models, with the two rows corresponding to opposite configurations.

symbols across enantiomers, failing to capture stereochemical differences. These results highlight the robustness and stereochemical sensitivity of ChiDeK in handling axial chirality. See Appendix E for more explanations and prediction examples.

**Rotation Analysis of Axial Chirality Representations.** To investigate how molecular representations generated by ChiDeK vary under rotation along the chiral axis, we systematically vary the torsion angle, with details provided in Appendix E.3. Figure 4 illustrates two representative examples. In both cases, the 18 conformers are partitioned into two opposite configurations (each comprising 9 conformers), consistent with the expected stereochemical symmetry. The trajectory reflects a transition from one configuration to its opposite and then back to the original, while the cosine similarity plots show high similarity within the same configuration and vice versa.

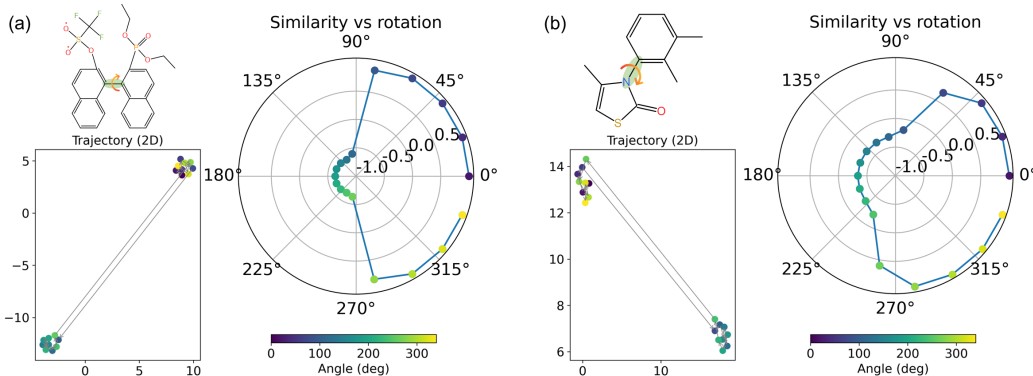

Figure 4: Visualization of how ChiDeK representations change under rotations along the chiral axis for two molecules. Each point in the trajectory plot obtained by UMAP (McInnes et al., 2018) represents the learned embedding at a given rotation angle, while each point in the polar plot denotes the cosine similarity between the embedding and the reference at degree 0.

## 5.2 ABLATION STUDIES

We perform ablation studies on the axial ECD prediction task, using the accuracy of peak height symbols as the primary evaluation metric. We examine strategies to address rank deficiency (QR and Reg denote QR decomposition and regularization, respectively), types of chiral encoders, and atom separation strategies (Sep.), as shown in Table 3. The ablation of missing or mislabeled chiral atoms

is presented in Appendix B.3. The comparison with ChiralFinder-enhanced model is presented in Appendix B.4.

Table 3: Ablation on model components.

| Model Components | | | Acc (%) ↑ |
|---|---|---|---|
| Chiral Encoder | Rank strategy | Separate | |
| Linear w/o. $M_{\mathrm{C}}$ | None | ✓ | $51.2 \pm 1.2$ |
| Linear w. $M_{\mathrm{C}}$ | None | ✓ | $68.7 \pm 0.6$ |
| Kernel-based | QR | ✗ | $69.8 \pm 0.4$ |
| Kernel-based | Reg | ✓ | $71.0 \pm 0.5$ |
| Kernel-based | None | ✓ | $68.3 \pm 0.6$ |
| Kernel-based | QR | ✓ | $71.2 \pm 0.6$ |

**(i) Strategies to Address Rank Deficiency.** Table 3 compares three approaches for handling rank deficiency: regularization (Reg), QR decomposition (QR), and no intervention (None). When handling rank deficiency, both Reg and QR outperform None, achieving similar performance.

**(ii) Choice of Chirality Encoder.** We examine alternative chiral encoder designs in Table 3. A linear encoder without access to $M_{\mathrm{C}}$ performs poorly (close to random guessing), indicating that it cannot capture chirality. In contrast, our kernel-based encoder yields a higher accuracy, confirming its ability to encode stereogenic features effectively.

**(iii) Separating Chiral Atom Types.** We test the effect of explicitly distinguishing chiral-related atoms ($\mathcal{I}_r$) from non-chiral atoms ($\mathcal{I}_n$). Models that separate chiral-related atoms and non-chiral atoms outperform those that treat them jointly, highlighting the importance of modeling their distinct roles within stereochemical environments.

## 6 CONCLUSION

We present ChiDeK (**Chi**ral **De**terminant **K**ernels), a unified framework for learning chiral molecular representations. By embedding the SE(3)-invariant chirality matrix through chiral determinant kernels and employing cross-attention between chiral and non-chiral atoms, ChiDeK explicitly captures both central and axial stereogenic features. Across multiple benchmarks, ChiDeK substantially outperforms baselines, particularly for axially chiral molecules. We further contribute a benchmark dataset for axial chirality, encompassing both ECD and OR prediction tasks. This dataset provides a foundation for future research in stereochemistry-aware machine learning. In future work, we plan to extend our unified stereochemical representation to additional forms of chirality, such as planar and helical chirality, and evaluate its utility in broader downstream applications, including docking-score prediction for chiral ligand-protein interactions and enantioselectivity prediction in asymmetric catalysis.

### ACKNOWLEDGMENTS

This work was supported by the National Key R&D Program of China (No. 2023YFC2811500) and the National Natural Science Foundation of China (No. 62272300).

### REPRODUCIBILITY STATEMENT

Code and data are available at `https://github.com/Meteor-han/ChiDeK`.

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

# A  DATASETS

Table 4: Summary of downstream datasets.

| Dataset | Chirality | Task | Size | Split |
|---|---|---|---|---|
| ChIRo (R/S) | Central | R/S classification | 466K conformers / 78K mols | 70/15/15 |
| ChIRo (Ranking) | Central | Enantiomer ranking | 335K conformers / 69K mols | 70/15/15 |
| CMCDS | Central | ECD prediction | 22,182 conformers / mols | 80/10/10 |
| ACMP | Axial | ECD and OR prediction | 1,192 conformers / mols | 80/10/10 |

## A.1  CENTRAL CHIRALITY

We adopt datasets from ChIRo (Adams et al., 2022): (i) R/S classification: the task is to classify the configuration (R or S) of a tetrahedral chiral center. The dataset is a subset of PubChem3D, comprising about 466K conformers of 78K enantiomers, each with exactly one tetrahedral chiral center. The same 70/15/15 splits are used. (ii) Enantiomer ranking: the task is to predict docking score rankings, evaluating whether the predicted relative ordering between enantiomers matches the ground truth. This dataset includes 335K conformers of 69K enantiomers (34.5K pairs), with enantiomers kept in the same data partition. We sample one conformer for each enantiomer in the batch during training. The same 70/15/15 splits are used.

In addition, we employ the CMCDS dataset for ECD prediction (Li et al., 2025), formulated as a multi-task, multi-label problem with three sub-tasks: predicting (1) the number of spectral peaks, (2) their positions, and (3) the sign of peak heights. CMCDS consists of ECD spectra for 22,190 chiral molecules (11,095 pairs), generated by large-scale theoretical calculations using Gaussian16 B.01 (Frisch et al., 2016b). After excluding four molecules without chiral centers, the dataset contains 22,182 conformers (11,091 pairs). We use random 80/10/10 splits while ensuring enantiomers are within the same partition.

## A.2  AXIAL CHIRALITY

We introduce a new benchmark, ACMP, for electronic circular dichroism (ECD) and optical rotation (OR) prediction. Specifically, we curate 650 axially chiral molecules from ChiralFinder (Shi et al., 2026) and generate initial 3D conformations using RDKit (Landrum, 2025). To refine these structures, a preliminary conformational optimization is first carried out using GFN2-xTB (Bannwarth et al., 2019), which is a semiempirical quantum mechanical method. Following this, the molecular geometries are fully optimized using the Gaussian 09 package (Frisch et al., 2016a), employing the B3LYP functional and the 6-31G(d) basis set.

For ECD, to obtain the spectral data, time-dependent DFT (TD-DFT) calculations are performed to compute the first 30 electronic excited states. These calculations use the long-range corrected CAM-B3LYP functional and the 6-311+G(d,p) basis set, yielding the excitation wavelengths and their corresponding rotatory strengths. ECD spectra curves are generated with Gaussian broadening (FWHM = 2/3), and an example is presented in Figure 5. We uniformly sample spectra at 1 nm intervals to produce sequence-form ECD representations. The labeling process follows the same procedure as CMCDS: peak positions are uniformly discretized into 20 classes, while peak heights are encoded using their sign (positive or negative). The number of peaks ranges from 0 to 6, and its distribution is illustrated in Figure 6. For OR, we compute the optical rotation at 589.3 nm using CAM-B3LYP/6-31G**. The resulting scalar rotation value is then mapped to a binary classification label, forming a two-class prediction task. All Gaussian calculations are conducted using 400 GB of memory with 192 CPU cores. To obtain enantiomeric counterparts, we generate opposite configurations by reflecting atomic coordinates across the z-axis. We finally get a dataset of 1,192 enantiomers (596 pairs). For evaluation, we adopt random 80/10/10 train/validation/test splits, ensuring that enantiomer pairs are kept within the same partition.

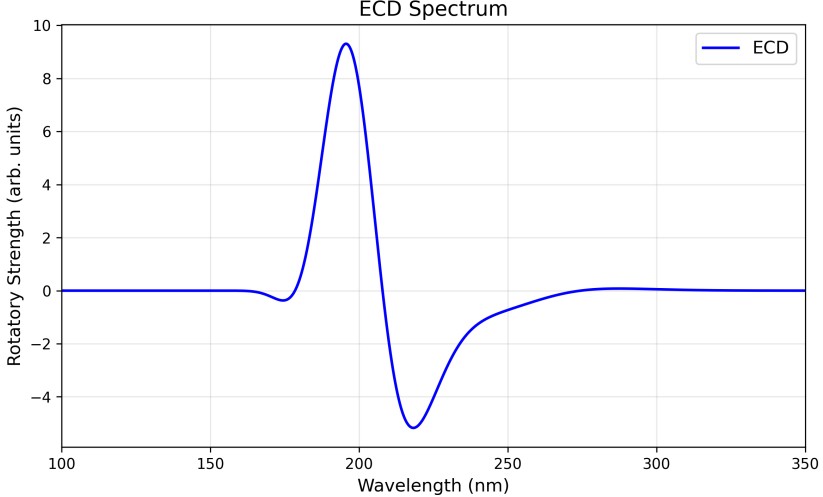

Figure 5: The ECD curve example for an axial chiral molecule.

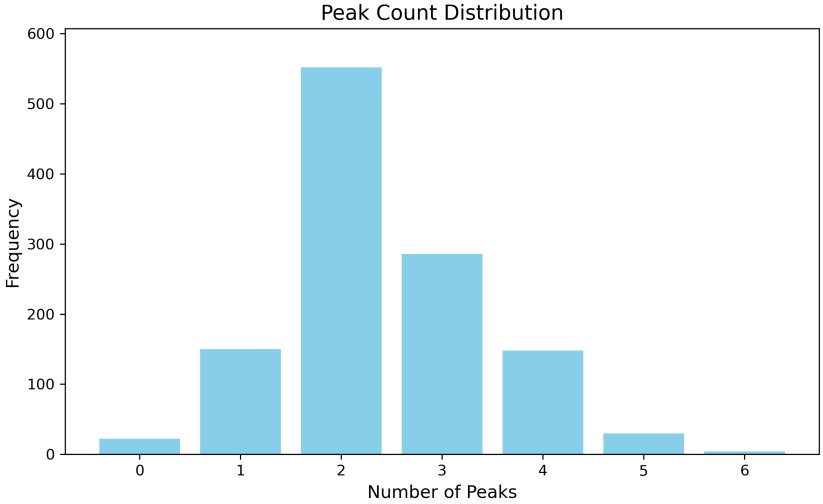

Figure 6: Distribution of ECD peak numbers in ACMP dataset.

# B   ADDITIONAL EXPERIMENTS

## B.1   FINE-GRAINED PERFORMANCE ON AXIAL-CHIRALITY SUBTYPES

To further investigate ChiDeK's performance across structural diversity, we provide a subtype-level breakdown on the test set, covering all seven axial-chirality subtypes. Table 5 reports detailed results for axial ECD-height prediction.

We observe that ChiDeK achieves strong performance on most subtypes ($\geq 70\%$), while Heterobiaryl (C–N) and Heterobiaryl (C–B) show lower performance ($\leq 60\%$), highlighting potential directions for future improvement.

## B.2   EVALUATION ON GENERAL MOLECULAR PROPERTY PREDICTION

To assess the broader applicability of ChiDeK beyond chirality-sensitive tasks, we evaluate the model on several moderate-sized MoleculeNet datasets Wu et al. (2018) where chirality is less

Table 5: Performance breakdown of seven subtypes for the axial ECD-height task.

| Subtype | # Enantiomers | # Height symbols | Acc (%) ↑ |
|---|---|---|---|
| Chiral atom pair | 10 | 26 | $71.5 \pm 1.8$ |
| Allene-like | 2 | 4 | $80.0 \pm 9.9$ |
| Spiral atom and chain | 4 | 8 | $90.0 \pm 5.0$ |
| Biaryl | 70 | 194 | $72.9 \pm 0.5$ |
| Heterobiaryl (C-N) | 12 | 20 | $57.0 \pm 4.0$ |
| Heterobiaryl (C-B) | 6 | 16 | $60.0 \pm 3.1$ |
| Nonbiaryl | 16 | 46 | $70.0 \pm 1.6$ |
| Total/Average | 120 | 314 | $71.2 \pm 0.6$ |

critical, including FreeSolv, BACE, BBBP, ClinTox, and SIDER. These benchmarks test general molecular property prediction rather than chiral discrimination.

As summarized in Table 6, ChiDeK achieves performance comparable to standard GNNs without pretraining (D-MPNN, Attentive FP). While it does not surpass highly optimized SOTA models on these general tasks, the results indicate that ChiDeK introduces no harmful inductive bias and retains broad representational capacity for generic molecular property prediction.

Table 6: Results on less chiral-related MoleculeNet datasets.

| Metric | ROC-AUC (%) ↑ | | | | RMSE ↓ |
|---|---|---|---|---|---|
| Datasets | **BBBP** | **BACE** | **ClinTox** | **SIDER** | **FreeSolv** |
| # Molecules | 2039 | 1513 | 1478 | 1427 | 642 |
| # Tasks | 1 | 1 | 2 | 27 | 1 |
| *W/o. pre-training* | | | | | |
| D-MPNN | $71.0 \pm 0.3$ | $80.9 \pm 0.6$ | $90.6 \pm 0.6$ | $57.0 \pm 0.7$ | $2.082 \pm 0.082$ |
| Attentive FP | $64.3 \pm 1.8$ | $78.4 \pm 0.0$ | $84.7 \pm 0.3$ | $60.6 \pm 3.2$ | $2.073 \pm 0.183$ |
| ChiDeK (Ours) | $68.9 \pm 0.8$ | $78.1 \pm 0.9$ | $79.2 \pm 1.1$ | $58.3 \pm 0.8$ | $2.412 \pm 0.109$ |
| *W. pre-training* | | | | | |
| PretrainGNN | $68.7 \pm 1.3$ | $84.5 \pm 0.7$ | $72.6 \pm 1.5$ | $62.7 \pm 0.8$ | $2.764 \pm 0.002$ |
| Uni-Mol | $72.9 \pm 0.6$ | $85.7 \pm 0.2$ | $91.9 \pm 1.8$ | $65.9 \pm 1.3$ | $1.480 \pm 0.048$ |

## B.3 ROBUSTNESS TO NOISY OR INCOMPLETE CHIRALITY ANNOTATIONS

To evaluate ChiDeK's robustness to imperfect chirality annotations, we conduct experiments on both central and axial chirality.

**Central chirality (R/S classification).** We systematically introduce noise by removing or mislabeling (1:1) chiral atoms in 2.5%, 5%, 7.5%, and 10% of molecules. For mislabeling, chiral centers are replaced with non-chiral atoms; for missing centers, the labels are simply removed. Table 7 shows a smooth and gradual degradation rather than a sharp collapse. With moderate levels of noise, the model maintains strong accuracy, demonstrating that the cross-attention architecture provides meaningful tolerance to incomplete or partially incorrect stereochemical inputs. These results confirm that ChiDeK is robust to moderate annotation errors, while also showing that reliable chiral identification naturally leads to stronger performance.

**Axial chirality (ECD-height prediction).** We evaluate the effect of removing ground-truth axial labels (with noise) on ECD-height prediction performance. In this setting, the average coverage (percentage of identified chiral atoms covering true chiral atoms) is 0.924, and the average IoU (intersection over union) between identified and true chiral atoms is 0.645. Similar to central chiral-

Table 7: Results of central R/S classification with different ratios of noise.

| Ratio | W/o. noise | 2.5% | 5% | 7.5% | 10% |
|---|---|---|---|---|---|
| ChiDeK | $99.8 \pm 0.1$ | $99.1 \pm 0.0$ | $98.2 \pm 0.0$ | $96.9 \pm 0.1$ | $96.1 \pm 0.2$ |

ity, removing ground-truth guidance reduces performance as shown in Table 8, demonstrating that reliable chiral identification is essential for optimal model performance.

Table 8: Results of axial ECD-height prediction with or without noise.

| Ratio | W/o. noise | W. noise |
|---|---|---|
| ChiDeK | $71.2 \pm 0.6$ | $65.8 \pm 0.4$ |

### B.4 COMPARISON WITH CHIRALFINDER-ENHANCED MODEL

We integrate the ChiralFinder-derived chirality matrix into ChIRo by concatenating the matrix features with the atomic embeddings (using zero vectors for non-chiral atoms) and evaluate this augmented model on the same axial-chirality benchmarks.

As shown in Table 9, the inclusion of the chirality matrix yields a clear improvement over the original ChIRo. Nevertheless, ChiDeK continues to outperform both variants. This result indicates that while the chirality matrix is beneficial, ChiDeK provides stronger representational capacity for modeling axial chirality.

Table 9: Comparison with ChIRo enhanced by the chirality matrix from ChiralFinder.

| Method | Rotation (%) | Symbol (%) |
|---|---|---|
| ChIRo | $50.0 \pm 0.1$ | $51.1 \pm 0.3$ |
| ChIRo + Chirality matrix | $62.1 \pm 0.3$ | $61.8 \pm 1.0$ |
| ChiDeK | $69.2 \pm 0.5$ | $71.2 \pm 0.6$ |

## C CHIRAL CROSS-ATTENTION

The learnable pairwise bias $p_{ij}^{(\ell)}$ is updated by:

$$p_{ij}^{(\ell+1)} = \frac{Q_i^{(\ell)} \left(K_j^{(\ell)}\right)^\top}{\sqrt{k}} + p_{ij}^{(\ell)}, \tag{19}$$

where $k$ is the hidden dimension. The attention outputs are then computed as

$$\text{Attention}\left(Q_i^{(\ell)}, K^{(\ell)}, V^{(\ell)}\right) = \text{softmax}\left(\frac{Q_i^{(\ell)}\left(K^{(\ell)}\right)^\top}{\sqrt{k}} + p_i^{(\ell-1)}\right) V^{(\ell)}. \tag{20}$$

Finally, the outputs of $L$ chiral cross-attention layers are aggregated by average pooling to get

$$H_{\text{global}} = \frac{1}{|\mathcal{I}_c|} \sum_{i \in \mathcal{I}_c} H_{c,i}^{(L)}. \tag{21}$$

This global representation serves as the input for downstream chirality property prediction.

# D   IMPLEMENTATION DETAILS

Whenever possible, baseline results are taken directly from the original publications; otherwise, we reproduce them using publicly available code. All results are reported over five test folds or five random runs.

## D.1   INITIALIZATION OF ATOMS

**Identification of Chiral and Chiral-related Atoms.**   For centrally chiral molecules, we use RDKit (Landrum, 2025) to detect chiral centers and their corresponding neighbors as chiral-related atoms. For axially chiral molecules, the chiral axis is labeled by chemists, while chiral-related atoms are determined using ChiralFinder (Shi et al., 2026). If ChiralFinder fails, we fall back to selecting up to four nearest neighbors as chiral-related atoms.

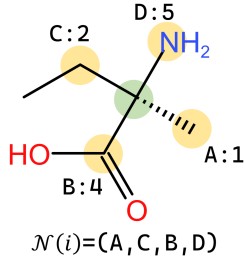

An example for ordering chiral-related atoms $\mathcal{N}(i)$ for atom $i$ is presented in Figure 7. After identifying the chiral atom (in green), suppose the indices of its chiral-related atoms (in orange) are A, B, C, and D, respectively. We use RDKit to calculate the CIP value for each atom (1, 4, 2, 5, respectively) and sort the indices in ascending order of their CIP priorities.

Figure 7: Ordering chiral-related atoms.

**Initial Atom Features.**   Each atom is initialized with a 52-dimensional feature vector, consistent with ChIRo (Adams et al., 2022), including atom type, degree, formal charge, number of hydrogens, and hybridization state. Chiral atoms are additionally associated with their chirality matrix, while non-chiral and chiral-related atoms share the same feature design. The embedding layers are implemented as multi-layer perceptrons.

## D.2   HYPERPARAMETER OPTIMIZATION

We tune hyperparameters via grid search (Table 10). Training epochs are set to 10 for R/S configuration classification, 200 for enantiomer ranking, and 50 for both central and axial ECD prediction. We use Adam optimizer (Kingma & Ba, 2014) and adopt a cosine learning rate schedule with a minimum learning rate of $0.1\times$ the initial value. As for the enantiomer ranking task, we follow ChIRo (Adams et al., 2022) and add the margin ranking loss.

Table 10: Hyperparameter search space for ChiDeK.

| Hyperparameter | Search Space |
|---|---|
| # Chiral Transformer Layers | $\{4, 8, 12, 16\}$ |
| # Chiral Transformer Heads | $\{2, 4, 8\}$ |
| Hidden Dimension | $\{64, 128, 256, 512\}$ |
| Projection Dimension | $\{32, 64, 128, 256, 512\}$ |
| Rank Strategy | $\{QR, Reg\}$ |
| Learning Rate | $\{$1e-4,5e-4$\}$ |
| Batch Size | $\{16, 32, 64, 128, 256\}$ |
| Weight of MarginRankingLoss | $\{0.1, 0.5, 1.0\}$ |

# E   ANALYSIS OF AXIAL ECD PREDICTION

## E.1   BASELINES FAIL TO LEARNING AXIAL CHIRALITY

Axial chirality presents unique challenges for ECD prediction. Our results demonstrate that ChiDeK achieves strong performance in this setting, primarily due to its explicit encoding of chirality through

the proposed chirality determinant kernels. By directly modeling stereochemically relevant features, ChiDeK is capable of distinguishing axial enantiomers and accurately predicting peak height symbols, a property that is essential for practical chirality-sensitive spectrum prediction.

By contrast, all baseline methods except SPMS perform poorly on axially chiral molecules, with each exhibiting specific limitations for distinct reasons, as analyzed below:

- **DimeNet++**: As an E(3)-invariant model that relies only on distances and angles, DimeNet++ is fundamentally incapable of distinguishing mirror-related structures. Its poor performance is therefore expected.

- **Tetra-DMPNN**: This model augments 2D GNNs with a central-chirality readout, but has no mechanism to capture axial chirality. As a result, it fails on this task.

- **SphereNet** and **ChIRo**: In principle, these models incorporate torsional information and should have some sensitivity to stereochemistry. However, axial chirality involves subtle global spatial arrangements that are not well captured by their architectures, leading to limited performance.

- **ECDFormer**: While this model introduces additional chiral encoding, it is designed for central chirality and lacks the necessary generalization to axial cases.

- **ChiGNN**: This method explicitly resolves atom permutations around tetrahedral centers, making it effective for central chirality. However, it does not generalize to axial stereogenic axes and thus completely fails in this setting.

### E.2 ECD PREDICTION EXAMPLES

Figure 8 presents two additional representative prediction comparisons. ChiDeK successfully assigns opposite peak-height symbols to opposite stereochemical configurations, while ChiGNN and Tetra-DMPNN (permutation variant) produce identical peak-height symbols across enantiomers, failing to capture stereochemical differences.

### E.3 DETAILS FOR ROTATING CHIRAL AXIS

We generate 18 conformers by rotating the torsion angle in 20-degree increments over 360 degrees. Each conformer corresponds to distinct coordinates, yielding different chirality matrices and thus different representations. We extract the embeddings before the predictor, apply UMAP (McInnes et al., 2018) for visualization, and record the trajectory as the torsion angle progresses from 0 to 340 degrees. Additionally, we calculate the cosine similarity between each embedding and the reference conformer at degree 0, and visualize the results using polar plots.

## F PROOF DETAILS

### F.1 CHIRALITY MATRIX

*Proof of Lemma 3.1.* Let $(\boldsymbol{z}_1, \boldsymbol{z}_2)$ be a pair of enantiomers that differ only in the stereochemical configuration of atom $i$. By definition of enantiomers there exists an isometry $T$ of $\mathbb{R}^3$ that maps the atomic coordinates of $\boldsymbol{z}_1$ to those of $\boldsymbol{z}_2$ and which is a composition of a rigid-body motion and a reflection; equivalently, one may write $T = R_2 \circ R_1$ with $R_1 \in \mathrm{SE}(3)$ and $R_2 \in O^-(3)$ (an orthogonal transformation with determinant $-1$). In particular, the local coordinate frame used to form the chirality matrix $\boldsymbol{M}_{\mathrm{C}}(i)$ for $\boldsymbol{z}_2$ is obtained from that for $\boldsymbol{z}_1$ by applying the linear part of $T$, which has determinant $-1$.

Since $i$ is a chiral atom, its substituent vectors (used to build $\boldsymbol{M}_{\mathrm{C}}(i)$) are non-coplanar, hence $\det\big(\boldsymbol{M}_{\mathrm{C}}(i)\big) \neq 0$. The sign of the determinant is well-defined.

Applying Proposition 3.1, first note that the rigid-body component $R_1 \in \mathrm{SE}(3)$ leaves the determinant invariant:

$$\det\big(\boldsymbol{M}_{\mathrm{C}}^{(\boldsymbol{z}_1)}(i)\big) = \det\big(R_1 \boldsymbol{M}_{\mathrm{C}}^{(\boldsymbol{z}_1)}(i)\big). \tag{22}$$

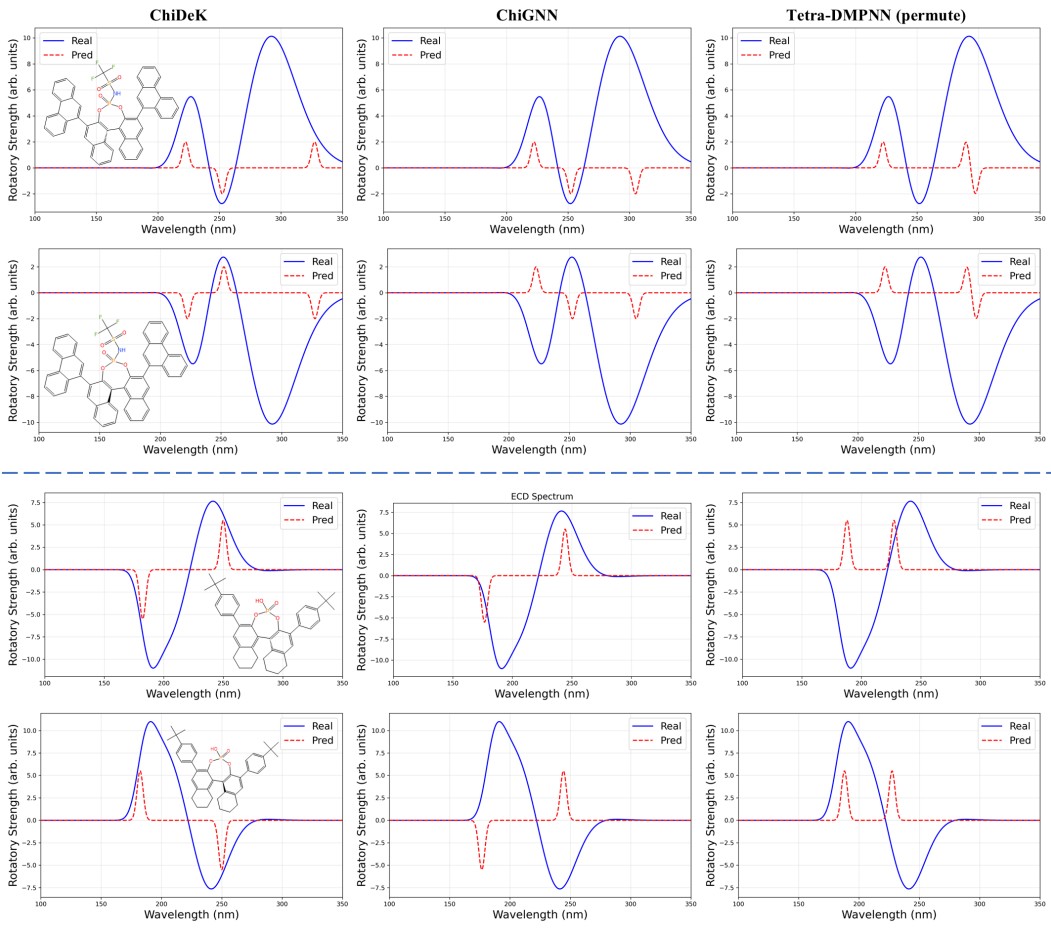

Figure 8: Axial ECD prediction examples of two pairs of enantiomers.

Then the reflection $R_2 \in O^-(3)$ flips the sign of the determinant:

$$\det\left(R_2 R_1 \boldsymbol{M}_{\mathrm{C}}^{(\boldsymbol{z}_1)}(i)\right) = -\det\left(R_1 \boldsymbol{M}_{\mathrm{C}}^{(\boldsymbol{z}_1)}(i)\right) = -\det\left(\boldsymbol{M}_{\mathrm{C}}^{(\boldsymbol{z}_1)}(i)\right). \tag{23}$$

$R_2 R_1 \boldsymbol{M}_{\mathrm{C}}^{(\boldsymbol{z}_1)}(i)$ is exactly the chirality matrix for atom $i$ in the enantiomer $\boldsymbol{z}_2$, i.e.,

$$\boldsymbol{M}_{\mathrm{C}}^{(\boldsymbol{z}_2)}(i) = R_2 R_1 \boldsymbol{M}_{\mathrm{C}}^{(\boldsymbol{z}_1)}(i). \tag{24}$$

Therefore,

$$\det\left(\boldsymbol{M}_{\mathrm{C}}^{(\boldsymbol{z}_2)}(i)\right) = -\det\left(\boldsymbol{M}_{\mathrm{C}}^{(\boldsymbol{z}_1)}(i)\right). \tag{25}$$

It follows that the determinants for the two enantiomers have opposite signs. By the sign convention in the lemma (positive determinant $\mapsto$ 'R', negative determinant $\mapsto$ 'S'), the stereochemical configuration of $i$ is 'R' when $\det(\boldsymbol{M}_{\mathrm{C}}(i)) > 0$ and 'S' when $\det(\boldsymbol{M}_{\mathrm{C}}(i)) < 0$. □

Figure 9 illustrates how reflection flips the sign of the volume (determinant), yielding an opposite configuration.

### F.2 CHIRAL DETERMINANT KERNEL

*Proof of Lemma 4.1.* Let $\boldsymbol{M} = \boldsymbol{M}_{\mathrm{C}}(i) \in \mathbb{R}^{3\times 3}$ and $\boldsymbol{W} \in \mathbb{R}^{d_p \times 3}$ be full column rank (so the rank is 3). Define $\boldsymbol{O} := \boldsymbol{W}\boldsymbol{M} \in \mathbb{R}^{d_p \times 3}$ and perform QR decomposition:

$$\boldsymbol{O} = \boldsymbol{Q}\boldsymbol{R}, \quad \boldsymbol{Q}^\top \boldsymbol{Q} = \boldsymbol{I}_3, \quad \boldsymbol{Q} \in \mathbb{R}^{d_p \times 3}, \quad \boldsymbol{R} \in \mathbb{R}^{3\times 3} \text{ upper-triangular.} \tag{26}$$

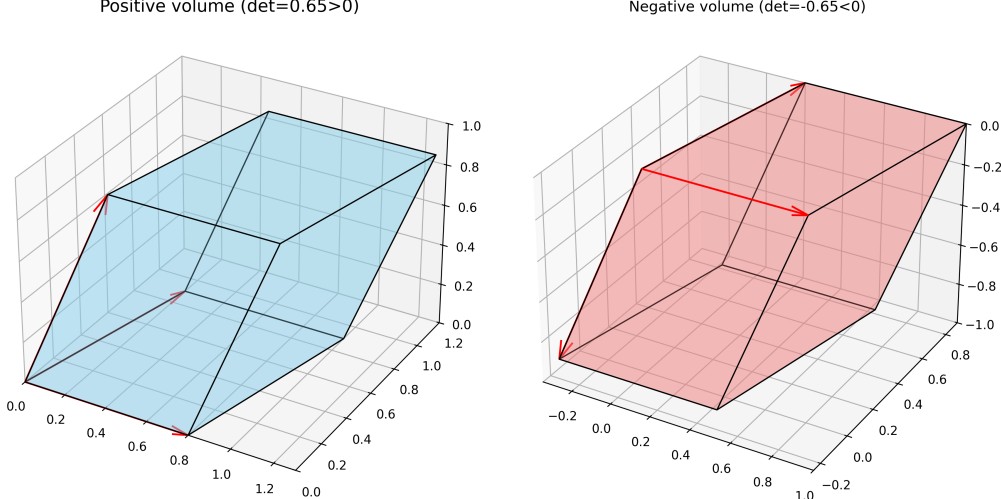

Figure 9: An example of reflection that flips the sign of volume.

Unlike the conventional QR convention, here we allow the diagonal entries of $\boldsymbol{R}$ to be negative, as is the case in numerical packages such as PyTorch. Consequently, the sign of $\det(\boldsymbol{R})$ is not predetermined but remains well defined.

Consider the Gram matrix

$$\boldsymbol{G} := \boldsymbol{W}^\top \boldsymbol{W}. \tag{27}$$

Since $\boldsymbol{W}$ has full column rank, $\boldsymbol{G} \succ 0$ and $\det(\boldsymbol{G}) > 0$. We have

$$\boldsymbol{R}^\top \boldsymbol{R} = \boldsymbol{O}^\top \boldsymbol{O} = (\boldsymbol{W}\boldsymbol{M})^\top (\boldsymbol{W}\boldsymbol{M}) = \boldsymbol{M}^\top \boldsymbol{W}^\top \boldsymbol{W}\boldsymbol{M} = \boldsymbol{M}^\top \boldsymbol{G}\boldsymbol{M}. \tag{28}$$

Taking determinants gives

$$\det(\boldsymbol{R})^2 = \det(\boldsymbol{M}^\top \boldsymbol{G}\boldsymbol{M}) = \det(\boldsymbol{M})^2 \det(\boldsymbol{G}). \tag{29}$$

Hence

$$|\det(\boldsymbol{R})| = |\det(\boldsymbol{M})| \cdot \sqrt{\det(\boldsymbol{G})}. \tag{30}$$

To recover the sign, observe that $\det(\boldsymbol{O}) = \det(\boldsymbol{Q})\det(\boldsymbol{R})$. Since $\boldsymbol{O} = \boldsymbol{W}\boldsymbol{M}$, this implies

$$\det(\boldsymbol{R}) = \frac{\det(\boldsymbol{W}\boldsymbol{M})}{\det(\boldsymbol{Q})}. \tag{31}$$

Here $\det(\boldsymbol{Q}) \in \{\pm 1\}$ because $\boldsymbol{Q}$ has orthonormal columns. Substituting gives

$$\det(\boldsymbol{R}) = \frac{\det(\boldsymbol{W})\det(\boldsymbol{M})}{\det(\boldsymbol{Q})}. \tag{32}$$

Combining with equation 30, the factor depending on $\boldsymbol{W}$ is

$$\alpha(\boldsymbol{W}) := \frac{|\det(\boldsymbol{W})|}{|\det(\boldsymbol{Q})|}\sqrt{\det\big((\boldsymbol{W}^\top \boldsymbol{W})/(\boldsymbol{W}^\top \boldsymbol{W})\big)}. \tag{33}$$

Since $\det(\boldsymbol{Q}) = \pm 1$, this ambiguity only flips the sign in a way consistent with $\det(\boldsymbol{M})$. Therefore, we can absorb it into the proportionality constant and write

$$\det(\boldsymbol{R}) = \sqrt{\det(\boldsymbol{G})}\det(\boldsymbol{M}) = \alpha(\boldsymbol{W})P_{\mathrm{C}}(i), \tag{34}$$

with $\alpha(\boldsymbol{W}) = \sqrt{\det(\boldsymbol{W}^\top \boldsymbol{W})} > 0$ independent of $\boldsymbol{M}$.

Finally, the invariance properties follow directly: if $\boldsymbol{M} \mapsto r\boldsymbol{M}$ for $r \in \mathrm{SO}(3)$, then $\det(\boldsymbol{M})$ is unchanged, hence $P_{\boldsymbol{W}}(i)$ is invariant. If $\boldsymbol{M} \mapsto s\boldsymbol{M}$ for $s \in O^-(3)$ with $\det(s) = -1$, then $\det(\boldsymbol{M})$ flips sign, hence $P_{\boldsymbol{W}}(i)$ flips sign as well. Thus $P_{\boldsymbol{W}}(i)$ inherits Proposition 3.1 and Lemma 3.1. $\square$

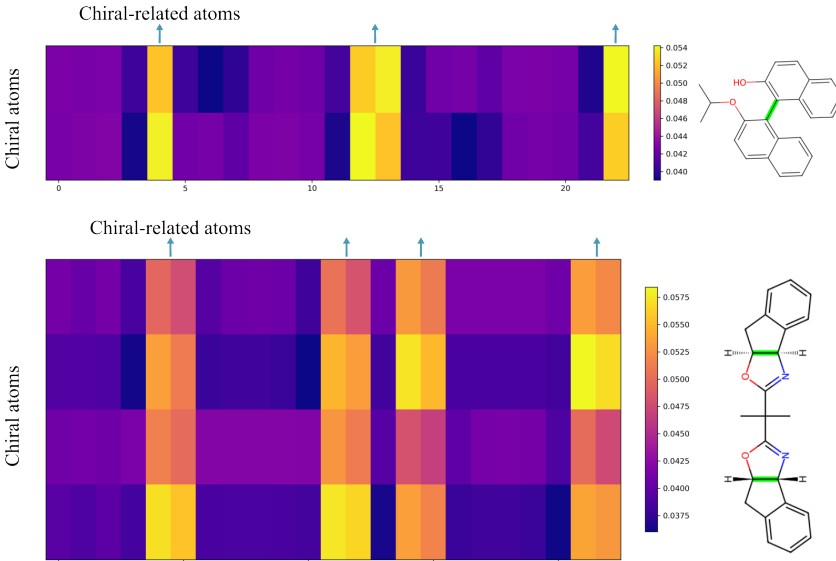

Figure 10: Two examples of cross attention weights. The chiral axis for each molecule is highlighted in green.

## G  VISUALIZATION OF THE CHIRAL TRANSFORMER

To better understand how ChiDeK leverages chiral information, we visualize the cross-attention weights between chiral atoms (used as queries) and all other atoms (used as keys and values) on the R/S classification task. Figure 10 illustrates two representative test examples from the axial ECD prediction task, depicting the head-averaged cross-attention weights in the final layer of the chiral transformer. In both cases, we observe that attention weights assigned to chiral-related atoms are higher than those assigned to non-chiral atoms. This demonstrates that ChiDeK successfully prioritizes stereochemically relevant atoms when integrating local chiral information into the global molecular representation. Conversely, non-chiral atoms receive comparatively lower weights, suggesting that the model effectively distinguishes their less critical role in stereospecific interactions.

