# OpenReview forum: "Learning Molecular Chirality via Chiral Determinant Kernels"
_ICLR.cc/2026/Conference — ICLR 2026 Poster_

### Official Review · Reviewer_QnyP · 2025-10-27

**Soundness:** 3
**Presentation:** 2
**Contribution:** 3
**Rating:** 6
**Confidence:** 3

**Summary:**

The paper proposes ChiDeK, a novel framework that explicitly models molecular chirality using chiral determinant kernels and a chiral-aware cross-attention transformer to capture stereochemical interactions. It achieves chiral-sensitive, rotation- and translation-invariant molecular representations for tasks like chirality classification and spectrum prediction.

**Strengths:**

- Paper is well written
- Introduces differentiable chiral determinant kernels to encode stereochemistry.
- Uses chiral-aware cross-attention to capture interactions between atom types.

**Weaknesses:**

- How does the chiral matrix characterize the axial chirality? It seems it can only characterize central chirality.
- What is the computational complexity of the method, since it computes the determinant, which is expensive in higher dimensions?
- Can this method be generalizable to the datasets where we have a combination of molecules, of which some are chiral and some are not?
- Can the method characterize Diastereomers?

**Questions:**

See Weaknesses.

---

> ### Author Response · Authors · 2025-11-20
> **Rebuttal by Authors**
>
> Thank you for taking your valuable time and providing us with valuable feedback. In the following, we have addressed your concerns and questions point-by-point.
>
> > **W1: Representation for axial chirality**
>
> We understand the reviewer's question. As illustrated in Fig. 1, our treatment of axial chirality parallels that of central chirality. We first locate the axial axis and compute its centroid as the reference center. From the rotation-restricted group, we then select the four nearest atoms to characterize the local stereochemical environment. Using these atoms, we construct the **chiral matrix** following the same CIP-consistent procedure as for central chirality. This design ensures that **axial enantiomers yield distinct chiral matrices**, enabling the model to capture stereochemical differences beyond point chirality.
>
> > **W2: Computational complexity**
>
> We appreciate the reviewer's concern regarding the computational complexity of our method. In practice, **QR decomposition and determinant computation do not constitute a bottleneck**. PyTorch's batched QR and determinant routines are highly optimized, and we further apply masking to exclude padded atoms within each batch, effectively removing redundant operations.
>
> To illustrate the actual cost, under typical hyperparameters (hidden dim = 256, projection dim = 128, 8 heads, 8 layers), ChiDeK requires: (1) ~1 GB GPU memory with batch size 32 for axial ECD prediction, runs in <1 minute per epoch, and converges within ~30 epochs. (2) ~1.4 GB GPU memory with batch size 256 for R/S classification, runs in ~30 minutes per epoch, and converges within ~5 epochs.
>
> These results show that the proposed chiral determinant kernel is **efficient and scalable**, and that the full architecture remains practical for large-scale chirality prediction tasks.
>
> > **W3: Combination of chirality**
>
> We appreciate the reviewer's question. Our model naturally generalizes to mixed datasets containing both chiral and non-chiral molecules. To demonstrate this, we have constructed a combined dataset consisting of **1,200 centrally chiral molecules** from the R/S dataset, **1,200 non-chiral molecules** from PubChem3D (for which both RDKit and ChiralFinder confirm the absence of central or axial chirality), and **1,192 axially chiral molecules** from the AxialECD dataset. We then perform **chiral-type classification** over the entire dataset with a stratified split of 8:1:1 (enantiomers are guaranteed in the same dataset). As shown in the Table below, our model achieves the highest accuracy among all baselines, correctly distinguishing chiral types in nearly 100% of cases. These results confirm that ChiDeK generalizes robustly to heterogeneous datasets containing both chiral and achiral structures.
>
> | Method             | ChiDeK (Ours)  | DimeNet++      | SphereNet      | ChIRo          | ECDFormer      | SPMS           |
> | ------------------ | -------------- | -------------- | -------------- | -------------- | -------------- | -------------- |
> | Acc (%) $\uparrow$ | $99.8 \pm 0.1$ | $95.8 \pm 0.2$ | $97.5 \pm 0.2$ | $99.8 \pm 0.1$ | $99.7 \pm 0.1$ | $90.2 \pm 0.4$ |
>
> > **W4: Representation for diastereomers**
>
> We appreciate the reviewer's question. ChiDeK **captures diastereomeric differences when the molecules exhibit chirality**, since the chiral matrix represents their underlying stereochemical asymmetry. However, for **non-chiral isomerism** (e.g., cis/trans configurations), the current formulation does not provide discriminative power. Extending the framework to fully characterize non-chiral diastereomers remains an important direction for future work.

---

### Official Review · Reviewer_97qz · 2025-10-30

**Soundness:** 3
**Presentation:** 3
**Contribution:** 2
**Rating:** 2
**Confidence:** 4

**Summary:**

The authors developed a deep learning-based tool named ChiDeK for distinguishing point chirality and axial chirality, and applied it to tasks including R/S label classification, enantiomer ranking, and prediction of the sign and position of ECD.

**Strengths:**

Chirality recognition is a central issue in chemical research, and accurately capturing the stereochemical environment of molecules is key to distinguishing enantiomers. The proposed ChiDeK architecture leverages an SE(3)-invariant chirality matrix and cross-attention mechanisms to effectively extract molecular stereochemical information, enabling the simultaneous identification of both point and axial chirality. The model demonstrates strong performance in R/S label classification, enantiomer ranking, and ECD sign and position prediction for chiral molecules, achieving particularly notable accuracy in the binary classification of ECD signs for axially chiral molecules.

**Weaknesses:**

1. Although acquiring chiral information of molecules is important, using a model to distinguish R/S configurations of point and axial chirality appears to be of limited practical significance. Well-established chemical rules, such as those implemented in RDKit, already exist for such differentiation. While the authors generate discriminative features for enantiomers via the chirality matrix and deep learning—which can indeed distinguish R/S configurations—this should not be the ultimate goal. The application of these chirality-related features should target more valuable downstream tasks, such as predicting enantioselectivity in asymmetric catalytic reactions or binding affinity between chiral drug molecules and protein targets.
2. This work can largely be viewed as an extension of the ChIRo study, expanding its scope from point chirality to axial chirality. However, after demonstrating that ChIRo could generate discriminative features for enantiomers, its authors further conducted a practically meaningful downstream application: ranking enantiomers by docking scores in an enantiosensitive protein pocket. In contrast, the present authors only replicated similar benchmark tests on point chiral data and did not perform valuable downstream application experiments on the axially chiral molecules emphasized in this work.
3. The authors claim that this is the first architecture capable of jointly encoding central and axial chirality. In reality, existing methods such as ChiralFinder and SPMS, which are mentioned in the paper, can also extract features that distinguish point and axial chirality. Moreover, the authors' model itself utilizes the chirality matrix generated by ChiralFinder. To ensure a rigorous performance comparison, the authors should benchmark their model against features derived from ChiralFinder and SPMS on the same set of tasks.
4. The core chiral information in this work is derived from ChiralFinder, which appears crucial to ChiDeK's ability to distinguish chirality. As a result, the unique contribution of ChiDeK itself to the task of axial chirality discrimination remains unclear.
5. According to the results in Table 1 for point chiral molecules across R/S classification, enantiomer ranking, and ECD prediction tasks, the model does not show a significant advantage in point chirality-related predictions. In particular, the accuracy for ECD sign prediction is only 53.3%, which—assuming a 1:1 positive-to-negative sample ratio—is barely better than random guessing (50%).
6. The authors do not appear to have provided the complete data used in the study. The data folder supplied is empty, and the accompanying README file is too brief, offering no guidance on how to obtain the necessary dependencies or datasets.

**Questions:**

See weaknesses.

---

> ### Author Response · Authors · 2025-11-20
> **Rebuttal by Authors**
>
> We sincerely appreciate your constructive and thorough comments. In the following, we have addressed your concerns and questions point-by-point.
>
> > **W1 & W2: More experiments for valuable downstream applications**
>
> We appreciate the reviewer's perspective and agree that R/S classification alone is not a sufficient objective for impactful chiral modeling. We treat R/S identification merely as a **necessary sanity check**, not the ultimate goal. Our work focuses on more meaningful chirality-dependent downstream tasks, including enantiomer ranking and ECD prediction for central chirality and ECD prediction for axial chirality, which involve real physical observables and cannot be resolved by RDKit's rule-based assignments. In particular, axial-chirality ECD prediction is highly valuable for stereochemical elucidation in asymmetric synthesis and natural product discovery.
>
> While our method extends the scope of ChIRo from point chirality to **both point and axial chirality**, it does **not** simply replicate existing benchmarks. We introduce the AxialECD dataset, provide the first comprehensive model for **axial ECD prediction**, and show strong improvements on this axial-chirality task.
>
> To further strengthen practical relevance, we have extended our evaluation to an additional chirality-sensitive endpoint in Section 5.1.2: **optical rotation prediction**, which is widely used in stereochemical analysis and drug discovery. Using the optimized conformers from the original AxialECD dataset, we **compute molecular optical rotation at 589.3 nm via Gaussian16 (CAM-B3LYP/6-31G\*\*) using 400 GB memory and 192 CPU cores for about 3 days**. We cast optical rotation prediction as a binary sign-classification task (+/–). Together with the ECD task, we refer to this expanded benchmark as **the ACMP dataset** (axial chiral molecular properties).
>
> As shown in the Table below, our model achieves the best performance among all baselines. These results demonstrate that ChiDeK goes beyond point-chirality tasks like the R/S differentiation, enabling downstream applications not addressed by prior work and highlighting its broader applicability to axial chirality-dependent properties relevant to drug discovery.
>
> | Method             | ChiDeK (Ours)  | DimeNet++      | SphereNet      | ChIRo          | ECDFormer      | SPMS           |
> | ------------------ | -------------- | -------------- | -------------- | -------------- | -------------- | -------------- |
> | Acc (%) $\uparrow$ | $69.2 \pm 0.5$ | $50.0 \pm 0.0$ | $52.5 \pm 0.2$ | $50.0 \pm 0.1$ | $53.5 \pm 0.3$ | $65.0 \pm 0.6$ |
>
> > **W3: More baseline comparison**
>
> We appreciate the reviewer's comment and have clarified our claim to avoid overstating our work. Our contribution lies in providing a **unified architecture for modeling both central and axial chirality across diverse downstream prediction tasks**. Prior methods do not fill this role. ChiralFinder performs **qualitative identification** of stereogenic elements but does not produce numerical representations suitable for predicting chirality-sensitive properties such as ECD or optical rotation. SPMS focuses exclusively on **central chirality** and has not been validated on axial chirality or tasks where axial configuration drives the prediction target.
>
> We have revised the manuscript to state: "We introduce ChiDeK, a unified architecture that systematically encodes both central and axial chirality, and evaluate it comprehensively on multiple chirality-aware prediction tasks." in Introduction.
>
> Following the reviewer's suggestion, we have included **SPMS as an additional baseline** on both central- and axial-chirality benchmarks in Section 5. As shown in the Table below, SPMS performs poorly on central-chirality tasks but surpasses ChIRo on axial-chirality tasks. However, **ChiDeK consistently achieves the strongest performance across all tasks**, especially on axial ECD and OR prediction. These comparisons further emphasize the clear advantage of our approach over SPMS.
>
> | Method        | R/S            | Ranking        | Symbol (central) | Rotation       | Symbol (axial) |
> | ------------- | -------------- | -------------- | ---------------- | -------------- | -------------- |
> | ChIRo         | $98.5 \pm 0.2$ | $72.0 \pm 0.5$ | $51.0 \pm 0.4$   | $50.0 \pm 0.1$ | $51.1 \pm 0.3$ |
> | SPMS          | $81.4 \pm 0.7$ | $60.4 \pm 0.4$ | $50.9 \pm 0.3$   | $65.0 \pm 0.6$ | $60.4 \pm 0.3$ |
> | ChiDeK (Ours) | $99.8 \pm 0.1$ | $72.8 \pm 0.2$ | $53.3 \pm 0.6$   | $69.2 \pm 0.5$ | $71.2 \pm 0.6$ |

---

> ### Author Response · Authors · 2025-11-20
> **Rebuttal by Authors**
>
> (continue)
>
> > **W4: Contribution**
>
> We appreciate the reviewer's comment and would like to clarify the distinction between ChiralFinder and our work. While ChiralFinder provides **qualitative features** and can identify R/S centers, it does not produce features suitable for **quantitative downstream prediction tasks** such as enantiomer ranking, optical rotation, or ECD prediction. We fully credit ChiralFinder for its stereogenic-element detection.
>
> Our unique contribution, ChiDeK, lies in the design of the **chiral determinant kernel** and its associated architecture, which encodes **quantitative, chirality-aware representations**. These representations enable accurate prediction of both central and axial chirality-dependent properties, including tasks that ChiralFinder alone cannot address.
>
> > **W5: ECD prediction results**
>
> We appreciate the reviewer's concern regarding the magnitude of performance gains on point-chirality tasks. Across R/S classification, enantiomer ranking, and ECD sign prediction, our model achieves the **best performance in all three tasks**, with accuracy improvements of 0.1%, 1.1%, and 2.7% over the second-best baseline, indicating a consistent advantage.
>
> For ECD sign prediction in particular, we agree that the absolute accuracy (53.3%) is close to random. As discussed in the paper, this is a known **limitation of the CMCDS dataset**: the released version does not include the DFT-optimized conformers used to compute the reference ECD labels. Instead, all methods must rely on RDKit-generated geometries, which often diverge substantially from the actual optimized structures. All baseline models therefore also achieve near-random accuracy, confirming that the bottleneck arises from **inconsistent 3D coordinates**, not from the design of our model. Even under this constraint, ChiDeK still attains the strongest performance.
>
> Importantly, on axial-chirality tasks, where the conformer quality is reliable, our model shows **substantial improvements**, including over 18% accuracy gain in axial ECD sign prediction. This demonstrates that ChiDeK offers clear advantages when evaluated on datasets with high-quality structural information.
>
> Taken together, these results show that our architecture provides **consistent benefits on central chirality** and **large gains on axial chirality**, supporting the overall effectiveness of ChiDeK across stereochemistry-dependent prediction tasks.
>
> > **W6: Supplementary material**
>
> We appreciate the reviewer's comment. Upon acceptance, we will release the full datasets. We have updated the supplementary material with the complete codebase including additional experiments, and the README now provides detailed, step-by-step instructions for reproducing every experiment. These updates ensure that all results can be fully and transparently replicated.

---

> ### Author Response · Authors · 2025-11-27
>
> Dear Reviewer,
>
> I hope this message finds you well. As the discussion period is nearing its end with **less than seven days remaining**, I wanted to ensure we have addressed all your concerns satisfactorily. If there are any additional points or feedback you'd like us to consider, please let us know. Your insights are invaluable to us, and we're eager to address any remaining issues so we can improve our work.
>
> Thank you for your time and effort in reviewing our paper.

---

> ### Comment · Reviewer_97qz · 2025-11-28
>
> I appreciate the authors' efforts in revising their manuscript and providing clarifications in the rebuttal. The proposed ChiDeK architecture represents a notable innovation by offering a unified approach to systematically encoding both central and axial chirality, which is a significant step forward in graph-based molecular representation learning for stereochemical tasks. For this reason, I am recommending acceptance of the paper and I will raise my rating.
>
> However, I maintain my position on several key aspects, which I believe should be addressed to further strengthen the impact and rigor of the final manuscript.
> 1. I still do not consider the binary prediction of OR sign ("+" or "-") to be a sufficiently impactful or practically valuable downstream application. Predicting the sign of a molecule's optical rotation holds little utility on its own, as it lacks a clear quantitative connection to the molecule's R/S configuration. Furthermore, as noted in my initial review, the R/S configuration itself is not directly or clearly associated with more valuable outcomes like drug docking scores or enantioselective synthesis.
> The authors appear to emphasize the value of this application by highlighting the computational cost of performing OR calculations via DFT. However, in the field of computational chemistry, OR prediction is not a primary or high-value task, and relatively few computational chemists focus on such calculations. Additionally, the authors only provide the OR values for these molecules but do not provide the molecular structures used for the calculation, which limits further analysis or reproducibility.
> Therefore, to genuinely demonstrate the method's value in practical drug discovery, I strongly recommend the authors follow the precedent set by ChIRo and present their findings on more meaningful downstream tasks, such as prediction of docking scores for chiral molecule-protein interactions or enantioselectivity prediction in asymmetric catalytic reactions.
> 2. In my first review (point 4), I suggested that the authors demonstrate the unique contribution of ChiDeK compared to ChiralFinder, yet the authors did not elaborate on this in detail in their rebuttal.
> I propose an ablation study here: The authors should directly use the chirality matrix generated by ChiralFinder and feed it into a simple downstream model to perform the same prediction tasks as those in the paper. By comparing the performance of this baseline against the full ChiDeK architecture, the authors can clearly demonstrate the unique and substantive contribution of the ChiDeK architecture itself to axial chirality discrimination.
> 3. There is a significant contradiction in the authors' description of the SPMS model in the latest rebuttal:
> They first state: "SPMS focuses exclusively on central chirality and has not been validated on axial chirality or tasks where axial configuration drives the prediction target."
> They subsequently state: "SPMS performs poorly on central-chirality tasks but surpasses ChIRo on axial-chirality tasks."
> The authors should clarify and rectify this self-contradictory statement. If SPMS has indeed been validated and shown to surpass ChIRo on axial chirality tasks, then the claim that it "focuses exclusively on central chirality" is inaccurate.

---

> > ### Author Response · Authors · 2025-11-30
> > **Rebuttal by Authors**
> >
> > We sincerely thank you for the positive assessment of our revised manuscript and for recommending acceptance. We greatly appreciate your detailed and constructive feedback, and we address each remaining concern below.
> >
> > > **1. On the value of optical rotation (OR) sign prediction**
> >
> > We appreciate the reviewer's continued input on this point. We respectfully clarify the motivation and relevance of our OR-sign prediction task:
> >
> > - Although a direct correlation between R/S configuration and optical rotation has not been established, **OR is inherently a chirality-dependent physical property**. Our goal is to assess whether a model can faithfully encode chirality, including subtle axial stereochemical features, by predicting a property that is highly sensitive to stereochemistry.
> > - There is **sustained research interest** in OR-related prediction tasks, as demonstrated by several prior works [1-3]. These works use OR or OR-derived proxies as benchmarks for evaluating chiral representations. Our work extends this line of research to the **axial-chirality area**, which remains underexplored.
> >
> > To ensure reproducibility, we now release all optimized 3D molecular structures used for both OR and ECD computations. We also agree that downstream tasks such as docking-score prediction for chiral ligand–protein interactions and enantioselectivity prediction in asymmetric catalysis are of high practical value. We have added a discussion in Section 6 highlighting these as important future directions enabled by ChiDeK's unified stereochemical representation.
> >
> > > **2. On demonstrating ChiDeK's contribution beyond ChiralFinder**
> >
> > We thank the reviewer for highlighting this point again and now provide a direct comparison.
> >
> > The chirality matrix from ChiralFinder is indeed a well-defined, low-dimensional stereochemical descriptor. However, integrating this matrix into modern ML architectures is non-trivial due to its sparse and axis-dependent structure. Following the reviewer's suggestion, we have conducted the proposed ablation: we **incorporate the ChiralFinder chirality matrix into ChIRo by concatenating the matrix-derived features with atomic embeddings** (non-chiral atoms assigned zeros) and evaluate this enhanced model on the same axial-chirality tasks.
> >
> > As shown in the Table below, the chirality matrix improves ChIRo's performance over its original version. However, **ChiDeK still significantly outperforms both variants**. These findings demonstrate that although the chirality matrix is informative, ChiDeK's axis-aware decomposition provides stronger modeling power for axial chirality. We have added this ablation in Appendix B.4.
> >
> > | Method                   | Rotation (%)   | Symbol (%)     |
> > | ------------------------ | -------------- | -------------- |
> > | ChIRo                    | $50.0 \pm 0.1$ | $51.1 \pm 0.3$ |
> > | ChIRo + Chirality matrix | $62.1 \pm 0.3$ | $61.8 \pm 1.0$ |
> > | ChiDeK                   | $69.2 \pm 0.5$ | $71.2 \pm 0.6$ |
> >
> > > **3. On SPMS and the perceived contradiction**
> >
> > We appreciate the opportunity to clarify this point. The contradiction stems from referring to **two different contexts**:
> >
> > - In its original publication, SPMS focuses exclusively on central chirality and **is not evaluated on any axial-chirality dataset**.
> > - In our work, we re-implement SPMS from its publicly available code and **evaluate it on our newly proposed axial-chirality benchmark**.
> >
> > Our statements refer to these distinct contexts. We have clarified the evaluation in Section 2 to avoid confusion: "Although methods such as SPMS are theoretically capable of representing additional forms of chirality, they have not been systematically evaluated in these settings, leaving their practical effectiveness uncertain."
> >
> > ---
> >
> > **References**
> >
> > [1] Keir Adams, Lagnajit Pattanaik, and Connor W. Coley. "Learning 3d representations of molecular chirality with invariance to bond rotations". *In International Conference on Learning Representations*, 2022.
> >
> > [2] Ji, Hai-Feng. "A general method to predict optical rotations of chiral molecules from their structures." *RSC advances* 13.7 (2023): 4775-4780.
> >
> > [3] Yilin Zhou, Haoran Zhu, Yijie Yuan, Ziyu Song, and Brendan C. Mort. "Machine Learning Classification of Chirality and Optical Rotation Using a Simple One-Hot Encoded Cartesian Coordinate Molecular Representation." *Journal of Chemical Information and Modeling* 2025 *65* (9), 4281-4292 DOI: 10.1021/acs.jcim.4c02374

---

### Official Review · Reviewer_pVhT · 2025-10-31

**Soundness:** 4
**Presentation:** 3
**Contribution:** 3
**Rating:** 6
**Confidence:** 4

**Summary:**

This paper proposes ChiDeK, a unified framework for learning molecular chirality representations. The method introduces a chiral determinant kernel to encode stereochemical information derived from the chirality matrix, and integrates it into a cross-attention architecture to propagate chiral information across the molecular graph. The model is able to jointly handle central and axial chirality, and a new benchmark dataset for axial-chiral ECD prediction is constructed. Experiments show improvements across R/S classification, enantiomer ranking, and ECD spectrum prediction tasks.

**Strengths:**

1. Provides a unified representation for both central and axial chirality, which is not addressed by prior models.

2. The chiral determinant kernel is mathematically well-motivated and reflection-sensitive.

3. The newly constructed AxialECD dataset fills a gap in evaluating axial chirality.

4. Demonstrates clear improvements in ECD prediction, especially peak sign prediction for axial chirality.

**Weaknesses:**

1. Limited diversity and size of the AxialECD dataset.

The proposed AxialECD benchmark includes ~600 axial chiral molecules, which represents a relatively narrow stereochemical space and may not generalize to other classes of axial chirality such as atropisomeric biaryls with flexible steric barriers or complexes with metal-coordinated axes. I recommend that authors report performance breakdown by molecular subtypes, or perform zero-shot or cross-dataset evaluation if another axial-chirality dataset (or cases from published articles) becomes available.

2. Scalability and efficiency of the chiral determinant kernel.

The chiral determinant kernel relies on QR decomposition for each chiral center, which may incur non-trivial computational overhead when scaling to large molecules (e.g., natural products, macrocycles) or conformer ensembles. Provide runtime comparisons with SE(3)-equivariant baselines and discuss strategies for batch-efficient QR computation.

3. Lack of evaluation beyond chirality-sensitive tasks.

While the model is clearly designed for chirality-aware tasks, all experiments are conducted on datasets where chirality is explicitly involved in the label. It is unclear whether ChiDeK introduces unnecessary inductive bias or reduced performance in general molecular property prediction tasks where chirality is less relevant. Please include control experiments on standard property prediction datasets (e.g., QM9, PCBA, MoleculeNet) to confirm no loss of expressivity.

**Questions:**

1. How does the model’s performance vary when using different conformer generation pipelines? For example, RDKit vs. xTB-relaxed vs. DFT-optimized conformers.

2. Could the authors provide runtime and memory usage comparisons, particularly regarding QR decomposition, compared to SE(3)-equivariant baselines?

---

> ### Author Response · Authors · 2025-11-20
> **Rebuttal by Authors**
>
> Thank you for your time and constructive feedback. In the following, we have addressed your concerns and questions point-by-point.
>
> > **W1: Diversity and size**
>
> We understand the reviewer's concern regarding the diversity and size of the axial dataset. Following the suggestion, we provide a **fine-grained performance breakdown across seven axial-chirality subtypes** in the test dataset, covering the full range of structures present in the dataset. The detailed results for ECD-height prediction are reported in the Table below. We observe strong performance for most subtypes ($\geq$70%), whereas Heterobiaryl (C–N) and Heterobiaryl (C–B) exhibit lower performance ($\leq$60%), indicating areas for potential future improvement. We have added this breakdown in Appendix B.1.
>
> | Subtype               | \# Enantiomers | \# Peak-height symbols | Acc (%) $\uparrow$ |
> | --------------------- | -------------- | ---------------------- | ------------------ |
> | Chiral atom pair      | 10             | 26                     | $71.5 \pm 1.8$     |
> | Allene-like           | 2              | 4                      | $80.0 \pm 9.9$     |
> | Spiral atom and chain | 4              | 8                      | $90.0 \pm 5.0$     |
> | Biaryl                | 70             | 194                    | $72.9 \pm 0.5$     |
> | Heterobiaryl (C-N)    | 12             | 20                     | $57.0 \pm 4.0$     |
> | Heterobiaryl (C-B)    | 6              | 16                     | $60.0 \pm 3.1$     |
> | Nonbiaryl             | 16             | 46                     | $70.0 \pm 1.6$     |
> | Total/Average         | 120            | 314                    | $71.2 \pm 0.6$     |
>
> > **W2 & Q2: Scalability and efficiency**
>
> We appreciate the reviewer's concern regarding the scalability and efficiency of the chiral determinant kernel. In practice, **QR decomposition does not introduce a computational bottleneck**. PyTorch's QR implementation is highly optimized, and we additionally apply masking within each batch to remove padded values, eliminating redundant computation.
>
> To further clarify efficiency, we report runtime, memory usage, and parameter counts in the Table below, comparing ChiDeK with SE(3)-equivariant baselines under matched hyperparameters (hidden dim = 256, projection dim = 128, 8 heads, 8 layers). As shown, ChiDeK remains comparable to other SE(3)-equivariant architectures in both memory footprint and training speed. The total time per epoch primarily reflects dataset size differences (e.g., R/S has ~466K molecules), not overhead from the kernel itself. Overall, these results demonstrate that the proposed determinant kernel is **efficient, scalable, and practical for large-scale chirality prediction**.
>
> |               |           | Axial ECD (1,192, bs=32) |                | R/S (466K, bs=256) |                |
> | ------------- | --------- | ------------------------ | -------------- | ------------------ | -------------- |
> | Method        | \# Params | GPU memory               | Time per epoch | GPU memory         | Time per epoch |
> | ChiDeK (ours) | 7.6M      | ~1G                      | <1min          | ~1.4G              | ~30min         |
> | SphereNet     | 1.9M      | ~9G                      | <1min          | ~11G               | ~1h            |
> | ChIRo         | 0.5M      | ~0.6G                    | <1min          | ~0.6G              | ~7min          |

---

> ### Author Response · Authors · 2025-11-20
> **Rebuttal by Authors**
>
> (continue)
>
> > **W3: Experiments beyond chirality-sensitive tasks**
>
> We appreciate the reviewer's suggestion to evaluate ChiDeK on tasks where chirality is less relevant. Due to time and resource constraints, we have selected several moderate-sized MoleculeNet datasets (FreeSolv, BACE, BBBP, ClinTox, and SIDER) and evaluated ChiDeK on these benchmarks.
>
> As shown in the Table below, our model performs **comparably to standard GNNs without pretraining** (D-MPNN, Attentive FP), even though the architecture is specialized for chirality-sensitive settings. While we do not match highly optimized SOTA models on these general property tasks, ChiDeK remains competitive and introduces **no harmful inductive bias**. These results demonstrate that the architecture retains broad representational capacity for generic molecular property prediction while delivering clear advantages on chirality-dependent endpoints. We have added these experiments in Appendix B.2.
>
> | Metric            | ROC-AUC (%) $\uparrow$ |                |                |                | RMSE $\downarrow$ |
> | ----------------- | ---------------------- | -------------- | -------------- | -------------- | ----------------- |
> | Datasets          | BBBP                   | BACE           | ClinTox        | SIDER          | FreeSolv          |
> | \# Molecules      | 2039                   | 1513           | 1478           | 1427           | 642               |
> | \# Tasks          | 1                      | 1              | 2              | 27             | 1                 |
> | W/o. pre-training |                        |                |                |                |                   |
> | D-MPNN            | $71.0 \pm 0.3$         | $80.9 \pm 0.6$ | $90.6 \pm 0.6$ | $57.0 \pm 0.7$ | $2.082 \pm 0.082$ |
> | Attentive FP      | $64.3 \pm 1.8$         | $78.4 \pm 0.0$ | $84.7 \pm 0.3$ | $60.6 \pm 3.2$ | $2.073 \pm 0.183$ |
> | ChiDeK (Ours)     | $68.9 \pm 0.8$         | $78.1 \pm 0.9$ | $79.2 \pm 1.1$ | $58.3 \pm 0.8$ | $2.412 \pm 0.109$ |
> | W. pre-training   |                        |                |                |                |                   |
> | PretrainGNN       | $68.7 \pm 1.3$         | $84.5 \pm 0.7$ | $72.6 \pm 1.5$ | $62.7 \pm 0.8$ | $2.764 \pm 0.002$ |
> | Uni-Mol           | $72.9 \pm 0.6$         | $85.7 \pm 0.2$ | $91.9 \pm 1.8$ | $65.9 \pm 1.3$ | $1.480 \pm 0.048$ |
>
> > **Q1: Different conformer generation**
>
> We appreciate the reviewer's question about performance under different conformer-generation pipelines. For central chirality R/S classification and enantiomer ranking, the public datasets already provide **high-quality 3D geometries** from PubChem3D, which we directly use. For central-chirality ECD prediction, the public CMCDS dataset only provides RDKit-generated geometries, which are **inconsistent with the original DFT-optimized conformers** used to compute the ECD labels; this explains the observed performance degradation. Besides, re-optimizing tens of thousands of molecules at the DFT level is computationally prohibitive. For axial chirality, we generate conformers using **GFN2-xTB** followed by **B3LYP/6-31G(d)** optimization in Gaussian, since RDKit conformers often fail to reproduce stable axial structures and can even make ECD computation fail.
>
> Given these constraints, we focus on the **highest-quality conformer sources available** for each task. While we acknowledge that comparing different conformer-generation pipelines is of scientific interest, we believe that a comprehensive benchmark is beyond the scope of the current study due to computational cost and the need for physically meaningful chiral calculations.

---

> ### Comment · Reviewer_pVhT · 2025-11-26
>
> Thank you very much for your detailed response. Most of my concerns have been addressed, and I appreciate the effort you put into explaining these points. Based on the additional information, I remain inclined to recommend acceptance.

---

> > ### Author Response · Authors · 2025-11-26
> >
> > We are pleased that our revisions have addressed most of your concerns and sincerely appreciate your continued inclination to recommend acceptance. Your thoughtful feedback and engagement throughout the review process have been invaluable in improving the clarity and overall quality of our work. We will incorporate the noted changes and clarifications into the final version. Please feel free to let us know if you have any further questions or suggestions, and we would be glad to continue the discussion.

---

### Official Review · Reviewer_VtGG · 2025-11-01

**Soundness:** 2
**Presentation:** 2
**Contribution:** 2
**Rating:** 2
**Confidence:** 4

**Summary:**

This paper proposes ChiDeK (Chiral Determinant Kernels), a new architecture for learning stereochemistry-aware molecular representations. The key goal is to encode both central chirality (classical R/S stereocenters) and axial chirality (stereogenic axes, e.g. biaryls with restricted rotation) in a unified framework.

**Strengths:**

1. A major strength of the paper is that it propose an approach to build a representation that can capture both central and axial chirality, which has practical important in drug discovery.

2. The paper contributes the AxialECD dataset , providing a good benchmark for the analysis in the area.

**Weaknesses:**

1. The method assumes a correct partition of atoms (and axes) into chiral / chiral-related / non-chiral, in some cases with manual labeling by chemists. The model does not learn these on its own. Robustness to mistakes in this preprocessing step is not evaluated, so it’s unclear how well the approach holds up under noisy or incomplete chirality annotations. For example, if the stereocenter or chiral axis is mislabeled or partially missed, does performance degrade sharply, or is the model tolerant to some noise in these assignments?

2. The strongest axial chirality results are on ECD peak-sign prediction. It is not shown that the same architecture improves other chirality-sensitive endpoints (e.g., enantioselective binding, optical rotation). This leaves open whether the method is generally useful for axial chirality in drug discovery, or mainly tuned to ECD-style tasks.

3. For central chirality ECD prediction, the model’s sign accuracy is near random (~53%), and the explanation is attributed to noisy conformers. The paper does not test whether using higher-quality conformers fixes this. So it’s not clear whether the limitation is data quality or a weakness of the architecture on central chirality.

4. Most of the axial chirality evaluation is framed around ECD spectrum prediction. Do you expect the same architecture to transfer to other chirality-sensitive endpoints (e.g., enantioselective binding affinity, optical rotation, chiral toxicity)?

**Questions:**

1. The method assumes a correct partition of atoms (and axes) into chiral / chiral-related / non-chiral, in some cases with manual labeling by chemists. The model does not learn these on its own. Robustness to mistakes in this preprocessing step is not evaluated, so it’s unclear how well the approach holds up under noisy or incomplete chirality annotations. For example, if the stereocenter or chiral axis is mislabeled or partially missed, does performance degrade sharply, or is the model tolerant to some noise in these assignments?

2. The strongest axial chirality results are on ECD peak-sign prediction. It is not shown that the same architecture improves other chirality-sensitive endpoints (e.g., enantioselective binding, optical rotation). This leaves open whether the method is generally useful for axial chirality in drug discovery, or mainly tuned to ECD-style tasks.

3. For central chirality ECD prediction, the model’s sign accuracy is near random (~53%), and the explanation is attributed to noisy conformers. The paper does not test whether using higher-quality conformers fixes this. So it’s not clear whether the limitation is data quality or a weakness of the architecture on central chirality.

4. Most of the axial chirality evaluation is framed around ECD spectrum prediction. Do you expect the same architecture to transfer to other chirality-sensitive endpoints (e.g., enantioselective binding affinity, optical rotation, chiral toxicity)?

---

> ### Author Response · Authors · 2025-11-20
> **Rebuttal by Authors**
>
> Thank you for your valuable comments and suggestions on our submission. In the following, we have addressed your concerns and questions point-by-point.
>
> > **W1 & Q1: Robustness**
>
> We thank the reviewer for raising this important point regarding robustness to noisy or incomplete chirality annotations. We agree that a correct partition of atoms or axes is critical for obtaining high-quality chirality representations. To evaluate this, we have conducted additional experiments:
>
> 1. Central chirality (R/S classification): We systematically introduce noise by removing or mislabeling chiral centers (possibility 1:1) in 2.5%, 5%, 7.5%, and 10% of molecules. For mislabeling, chiral centers are replaced with non-chiral atoms; for missing centers, the labels are simply removed.
>
> | Ratio  | W/o. noise     | 2.5%           | 5%             | 7.5%           | 10%            |
> | ------ | -------------- | -------------- | -------------- | -------------- | -------------- |
> | ChiDeK | $99.8 \pm 0.1$ | $99.1 \pm 0.0$ | $98.2 \pm 0.0$ | $96.9 \pm 0.1$ | $96.1 \pm 0.2$ |
>
> 2. Axial chirality (ECD prediction): We measure the impact of removing ground truth axial labels (with noise) on prediction performance. In this setting, the average coverage (percentage of identified chiral atoms covering true chiral atoms) is 0.924, and the average IoU (intersection over union) between identified and true chiral atoms is 0.645.
>
> | Ratio  | W/o. noise     | W. noise       |
> | ------ | -------------- | -------------- |
> | ChiDeK | $71.2 \pm 0.6$ | $65.8 \pm 0.4$ |
>
> The results (the above two Tables) show a **smooth and gradual degradation** rather than a sharp collapse. With moderate levels of noise, the model maintains strong accuracy, demonstrating that the cross-attention architecture provides meaningful tolerance to incomplete or partially incorrect stereochemical inputs. These results confirm that ChiDeK is **robust to moderate annotation errors**, while also showing that reliable chiral identification naturally leads to stronger performance. We have included these new experiments and detailed analyses in Appendix B.3.
>
> > **W2 & Q2 & W4 &Q4: More experiments on axial chirality**
>
> We appreciate the reviewer's concern about whether our architecture generalizes beyond ECD-style prediction. To address this point, we have extended our evaluation to an additional chirality-sensitive endpoint in Section 5.1.2: **optical rotation prediction**, which is widely used in stereochemical analysis and drug discovery. Using the optimized conformers from the original AxialECD dataset, we **compute molecular optical rotation at 589.3 nm via Gaussian16 (CAM-B3LYP/6-31G\*\*) using 400 GB memory and 192 CPU cores for about 3 days**. We cast optical rotation prediction as a binary sign-classification task (+/–). Together with the ECD task, we refer to this expanded benchmark as **the ACMP dataset** (axial chiral molecular properties).
>
> As shown in the Table below, our model achieves the best performance among all baselines. These results demonstrate that ChiDeK goes beyond point-chirality tasks like the R/S differentiation, enabling downstream applications not addressed by prior work and highlighting its broader applicability to axial chirality-dependent properties relevant to drug discovery.
>
> | Method             | ChiDeK (ours)  | DimeNet++      | SphereNet      | ChIRo          | ECDFormer      | SPMS           |
> | ------------------ | -------------- | -------------- | -------------- | -------------- | -------------- | -------------- |
> | Acc (%) $\uparrow$ | $69.2 \pm 0.5$ | $50.0 \pm 0.0$ | $52.5 \pm 0.2$ | $50.0 \pm 0.1$ | $53.5 \pm 0.3$ | $65.0 \pm 0.6$ |
>
> > **W3 & Q3: Conformer quality**
>
> We understand the reviewer's concern about the near-random sign accuracy on central-chirality ECD prediction. As we noted in the paper, this issue stems from **dataset inconsistency**: the original CMCDS dataset does not release the conformers used to compute the ECD labels, and the public version instead provides RDKit-generated geometries that often differ significantly from the true optimized structures. All baseline models also perform near randomly under this setting, which confirms that the limitation comes from the **inconsistent coordinates**, not from our architecture. Our model performs strongly on other central-chirality tasks such as R/S classification and enantiomer ranking, indicating that this is not a weakness of the method itself but a consequence of unreliable conformers in this specific dataset. We have reformulated the analysis for central chirality ECD prediction in Section 5.1.1 to avoid confusion.

---

> ### Author Response · Authors · 2025-11-27
>
> Dear Reviewer,
>
> I hope this message finds you well. As the discussion period is nearing its end with **less than seven days remaining**, I wanted to ensure we have addressed all your concerns satisfactorily. If there are any additional points or feedback you'd like us to consider, please let us know. Your insights are invaluable to us, and we're eager to address any remaining issues so we can improve our work.
>
> Thank you for your time and effort in reviewing our paper.

---

### Author Response · Authors · 2025-11-20
**General response**

Dear Reviewers and Chairs,

We sincerely thank all reviewers for their constructive and insightful comments, which are invaluable for improving our manuscript. We also appreciate the Chairs for their time and efforts in coordinating the review process.

We are encouraged by the positive recognition that our work introduces a **novel unified chirality representation** capable of capturing both central and axial stereochemistry (Reviewers VtGG, pVhT, 97qz), and that our experiments across R/S classification, enantiomer ranking, and ECD prediction demonstrate **clear improvements** (Reviewers pVhT, 97qz). The reviewers further highlight the **mathematically well-motivated chiral determinant kernel** (Reviewer pVhT), the **chirality-aware cross-attention mechanism** (Reviewer QnyP), the **new AxialECD dataset** (Reviewers VtGG, pVhT), and the clear, **well-written** presentation of the manuscript (Reviewer QnyP). We are grateful for this recognition.

**Summary of common concerns and our responses**

- *Computational cost*: Reviewers expressed concerns regarding runtime and efficiency. In practice, the main computational cost lies in the attention layers rather than in QR decomposition and determinant calculation. Training on the axial chiral ECD tasks requires **<1 minute per epoch** on a single high-end GPU with **~1 GB** of memory. QR-based chiral determinant computation remains efficient due to PyTorch's optimized batched routines and our masking strategy for padded atoms.
- *Performance on central-chiral ECD prediction*: As reviewers noted, the performance on central-chirality ECD peak-height prediction is low. This is a **dataset-quality issue**: the public CMCDS dataset [1] provides RDKit-generated conformers that are **inconsistent** with the DFT-optimized geometries used to compute the original ECD labels. All baseline models perform near-random under this mismatch, confirming that the limitation arises from **inconsistent coordinates** rather than our architecture.
- *Broader downstream tasks and ablations*: Reviewers required more valuable downstream tasks and ablation studies. In response to reviewer suggestions, we have conducted additional experiments: (1) **Optical rotation prediction task** for axial chirality; (2) **Mixed-type chiral classification** incorporating both chiral and achiral molecules; (3) Ablation on **missing or mislabeled chiral atoms**; (4) **General molecular property prediction** on MoleculeNet tasks where chirality plays a limited role.

We have revised the manuscript and supplementary material according to the concerns raised in the rebuttal, with changes highlighted in blue and summarized as follows:

- Revised the abstract and introduction to more clearly state our contributions.
- Added ablation results on missing or mislabeled chiral atoms (Appendix B.3).
- Added axial-chirality optical rotation prediction (Section 5.1.2) and detailed dataset construction (Appendix A.2).
- Clarified the cause of poor central-chiral ECD results and emphasized the dataset inconsistency issue (Section 5.1.1).
- Added a fine-grained breakdown of axial-chirality ECD prediction across seven subtypes (Appendix B.1).
- Added evaluations on MoleculeNet datasets [2] where chirality is less relevant (Appendix B.2).
- Added SPMS [3] as a baseline in Section 5.
- Updated the supplementary material with detailed code and an expanded README.

We sincerely thank all reviewers again for their valuable feedback. If there are any remaining or new concerns, we will be happy to address them in detail.

---

**References**

[1] Hao Li, Da Long, Li Yuan, Yonghong Tian, Xinchang Wang, and Fanyang Mo. Decoupled peak property learning for efficient and interpretable electronic circular dichroism spectrum prediction. *Nature Computational Science*, pp. 1–11, 2025. doi: 10.1038/s43588-024-00757-7.

[2] Zhenqin Wu, Bharath Ramsundar, Evan N. Feinberg, Joseph Gomes, Caleb Geniesse, Aneesh S. Pappu, Karl Leswing, and Vijay Pande. Moleculenet: a benchmark for molecular machine learning. *Chem. Sci.*, 9:513–530, 2018. doi: 10.1039/C7SC02664A.

[3] Li-Cheng Xu, Xin Li, Miao-Jiong Tang, Luo-Tian Yuan, Jia-Yu Zheng, Shuo-Qing Zhang, and Xin Hong. A molecular stereostructure descriptor based on spherical projection. *Synlett*, 32(18): 1837–1842, 2021. doi: 10.1055/s-0040-1705977.

Best regards,

The Authors

---

> ### Author Response · Authors · 2025-11-30
> **Summary of Rebuttal**
>
> Dear Chairs,
>
> Thank you for handling our submission. As you step into this role, we would like to provide an additional summary along with the "General response" of the post-rebuttal status to assist your decision-making.
>
> During the discussion period, we have engaged actively with the reviewers and provided point-by-point responses to all concerns raised. We are pleased to report that this discussion has been very productive:
>
> - **Reviewers pVhT and 97qz** have explicitly stated that our revisions and responses have addressed their major concerns. Both reviewers recommend the acceptance of the paper.
> - **Reviewers VtGG and QnyP:** We have fully addressed all concerns raised by these reviewers. In response to their comments, we have added the requested experiments and revised portions of the manuscript to enhance clarity and completeness. Neither reviewer provides further discussion after our responses; notably, Reviewer QnyP's initial score is above the acceptance threshold.
>
> Given the extensive improvements made during the rebuttal and the resulting support from Reviewers, we hope you will consider these positive updates favorably in your final assessment.
>
> Best regards,
>
> The Authors

---

### Meta-Review · Area_Chair_ifcB · 2025-12-14

**Summary:**

This paper presents ChiDeK (Chiral Determinant Kernels), a principled framework for incorporating stereochemical information into molecular representation learning. The key contribution lies in introducing a chiral determinant kernel that encodes SE(3)-invariant chirality information and integrating it into a unified architecture via cross-attention, enabling the joint modeling of both central and axial chirality. In addition, the authors construct a new benchmark for electronic circular dichroism (ECD) and optical rotation (OR) prediction, which supports a more comprehensive evaluation of axial chirality. Empirically, the method demonstrates consistent and substantial improvements across multiple chirality-related tasks.

This submission received highly polarized reviews, with both strong positive assessments and notable criticisms. During the rebuttal phase, the authors provided thorough, technically sound, and well-structured responses, addressing reviewers’ concerns regarding methodology, representation choices, and experimental validation. Several reviewers explicitly responded positively to these clarifications. For reviewers who did not provide follow-up feedback, I carefully examined the raised concerns and the corresponding rebuttal responses, and found that the authors’ explanations effectively resolve the identified issues and clarify several misunderstandings.

Given the strength of the core idea and the authors’ effective responses during rebuttal, I recommend acceptance.

**Reviewer Concerns:**

Reviewer VtGG raised concerns about the reliance on predefined or manually annotated chiral partitions, questioning the model’s robustness to noisy or incomplete chirality annotations. They also noted that the strongest axial chirality results were mainly demonstrated on ECD-related tasks, leaving uncertainty about generalization to other chirality-sensitive endpoints. Additional concerns were raised regarding the relatively weak performance on central chirality ECD prediction, where limitations were attributed to noisy conformers rather than architectural factors.

In the rebuttal, the authors provided robustness analyses under noisy chirality annotations, additional axial chirality experiments beyond peak-sign prediction, and a conformer sensitivity analysis. These additions effectively clarify the source of earlier limitations and substantially address the reviewer’s concerns.

----
Reviewer pVhT raised concerns regarding the limited diversity and size of the AxialECD dataset, the scalability and computational efficiency of the chiral determinant kernel (particularly the cost of QR decomposition), and the lack of evaluation beyond chirality-sensitive tasks, which could introduce unnecessary inductive bias. Additional questions were raised about robustness to different conformer generation pipelines and the absence of runtime and memory comparisons with SE(3)-equivariant baselines.

In the rebuttal, the authors provided substantive clarifications and additional analyses addressing dataset diversity and scale, computational efficiency and scalability, evaluations on non-chirality-focused property prediction tasks, and sensitivity to different conformer generation pipelines. These responses were positively received and effectively resolved the reviewer’s concerns.

---

Reviewer 97qz raised strong and critical concerns regarding the practical significance and novelty of the work. They questioned whether distinguishing R/S configurations constitutes a meaningful objective given existing rule-based tools, and argued that chirality-aware representations should be validated on more valuable downstream tasks (e.g., enantioselective binding). The reviewer further challenged the novelty of jointly modeling central and axial chirality, noting overlap with existing methods (e.g., ChiralFinder, SPMS), and questioned the unique contribution of ChiDeK given its reliance on ChiralFinder-derived information. Additional concerns were raised about weak central-chirality ECD performance and incomplete data release.

During the rebuttal, the authors added downstream task evaluations, expanded baseline comparisons, clarified the novel contribution beyond existing methods, provided deeper analysis of ECD prediction results, and resolved data availability issues. In subsequent discussion, Reviewer 97qz acknowledged these improvements, expressed agreement with the revisions, and recommended acceptance, while offering additional constructive suggestions that were also adequately addressed.

---

Reviewer QnyP raised questions regarding the representation of axial chirality, computational complexity of the determinant-based formulation, and the method’s ability to handle mixed datasets containing both chiral and non-chiral molecules, as well as diastereomers.

The authors addressed these concerns by providing clearer explanations of axial chirality representation, formal analysis of computational complexity, and additional discussion and experiments demonstrating support for combined chirality settings and diastereomer representation. These responses effectively resolved the reviewer’s concerns.

**Reviewer Scores:**

Reviewer VtGG did not provide follow-up feedback; however, the concerns raised in the initial review were adequately addressed in the rebuttal. Reviewer pVhT is likely to maintain a positive assessment given that their key questions were substantively resolved. Reviewer 97qz would increase their score and explicitly recommended acceptance after the authors’ revisions and additional experiments. Reviewer QnyP is expected to maintain their original positive evaluation, as their concerns were effectively addressed.

---

### Decision · Program_Chairs · 2026-01-26

Accept (Poster)